# Transferable enantioselectivity models from sparse data

Simone Gallarati[1,2,3], Erin M. Bucci[2,3], Abigail G. Doyle[2✉] & Matthew S. Sigman[1✉]

Identifying a catalyst class to optimize the enantioselectivity of a new reaction, either involving a different combination of known substrate types or an entirely unfamiliar class of compounds, is a formidable challenge. Statistical models trained on a reported set of reactions can help predict out-of-sample transformations[1–5] but often face two challenges: (1) only sparse data that offer limited information on catalyst–substrate interactions are available; and (2) simple stereoelectronic parameters may fail to describe mechanistically complex transformations[6,7]. Here we report a descriptor generation strategy that accounts for changes in the enantiodetermining step with catalyst or substrate identity, allowing us to model reactions involving distinct ligand and substrate types. As validating case studies, we collected data on enantioselective nickel-catalysed $C(sp^3)$ couplings[8] and trained statistical models with features extracted from the transition states and intermediates proposed to be involved in asymmetric induction. These models allow the optimization of poorly performing examples reported in a substrate scope and are applicable to unseen ligands and reaction partners. This approach offers the opportunity to streamline catalyst and reaction development, quantitatively transferring knowledge learned on sparse data to chemical spaces.

The interplay of experimental and computational chemistry has been indispensable in determining the origin of enantioselectivity in a myriad of asymmetric catalytic transformations[9] (Fig. 1a). This is especially the case in complex systems where interpretation of mechanistic experiments is challenging, thereby requiring quantum chemical methods for the analysis of energetically feasible competing pathways to ascertain the factors determining stereoinduction. However, reaction optimization from first-principle computations remains an outstanding goal as small errors in calculated energy differences between stereocontrolling transition states (TSs) lead to significant errors in stereochemical outcome predictions[10]. This problem is further exacerbated by the use of flexible catalysts that can access many thermally accessible conformations and the multitude of competing or off-cycle pathways in catalytic systems. The requirement for in-depth mechanistic knowledge at the outset of an investigation and high computational cost has therefore limited the applicability of density functional theory (DFT)-based virtual screening approaches to a few successful examples[11].

By contrast, multivariate linear regression (MLR) methods that derive predictive correlations between empirical data, such as product selectivity, and computed molecular descriptors of reaction components, especially catalyst structures, have become a versatile tool for reaction optimization, even in the absence of mechanistic hypotheses[12] (Fig. 1a). Using a comprehensive library of calculated catalyst features[13], MLR enables rapid virtual screening, effectively reducing time and costs associated with de novo ligand synthesis and testing. An essential step in this process is identifying relevant molecular features that quantitatively describe enantioselectivity while also serving as

mechanistic probes. To limit the computational cost of curating virtual libraries, parameters are often extracted from simplified metal–ligand complexes or free ligand surrogates to simulate the more complex catalytic intermediates (Fig. 1b). However, it has been demonstrated that sourcing descriptors from structures that more closely resemble relevant catalytic intermediates leads to improvements in a model's predictive ability[14–16]. Although such descriptors are highly transferable[17,18], they do not explicitly describe catalyst–substrate interactions, which instead are captured in DFT-optimized TS structures. However, performing high-level computations on a large scale is computationally prohibitive, and descriptors for MLR analysis are seldom sourced from DFT-optimized catalytic intermediates and TSs. As an alternative approach, we have shown that inclusive models may be trained via mechanism-specific parameterization of individual reaction components (reagents, catalyst, solvent and additives, whose separate features are concatenated into one reaction representation), allowing for the prediction of unseen ligand scaffolds and new substrate types[1,2,19]. Unfortunately, this approach is 'data hungry' and ill-suited for the low-data regime encountered in early reaction optimization campaigns. Furthermore, it cannot easily describe mechanistically complex transformations where the enantiodetermining step may change as the identity of the ligand or substrate changes. Consequently, incorporating catalyst and substrate structural features into a single model that allows chemical observations to be quantitatively transferred from one reaction to another remains a considerable challenge.

As a prominent example, developing enantioselective nickel (Ni)-catalysed $C(sp^3)$ cross-coupling reactions, which have emerged

[1]Department of Chemistry, University of Utah, Salt Lake City, UT, USA. [2]Department of Chemistry and Biochemistry, University of California, Los Angeles, Los Angeles, CA, USA. [3]These authors contributed equally: Simone Gallarati, Erin M. Bucci. ✉e-mail: abigaildoyle@g.ucla.edu; matt.sigman@utah.edu

**a** Computational approaches for understanding and predicting enantioselectivity in asymmetric catalysis

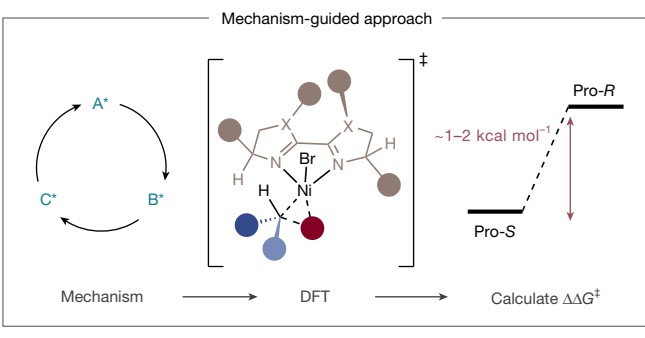
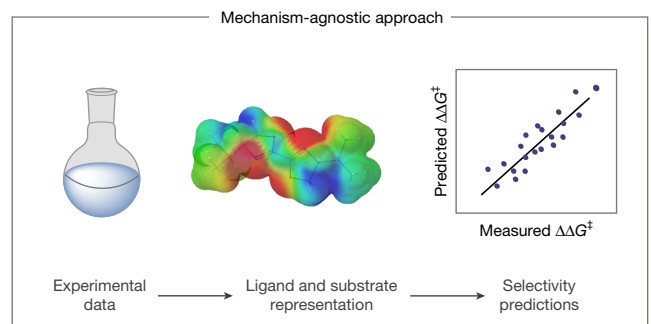

Mechanism-guided approach

Mechanism $\longrightarrow$ DFT $\longrightarrow$ Calculate $\Delta\Delta G^{\ddagger}$

~1–2 kcal mol$^{-1}$

Pro-*R*

Pro-*S*

Mechanism-agnostic approach

Experimental data $\longrightarrow$ Ligand and substrate representation $\longrightarrow$ Selectivity predictions

Predicted $\Delta\Delta G^{\ddagger}$ / Measured $\Delta\Delta G^{\ddagger}$

**b** MLR for asymmetric Ni-catalysed XEC

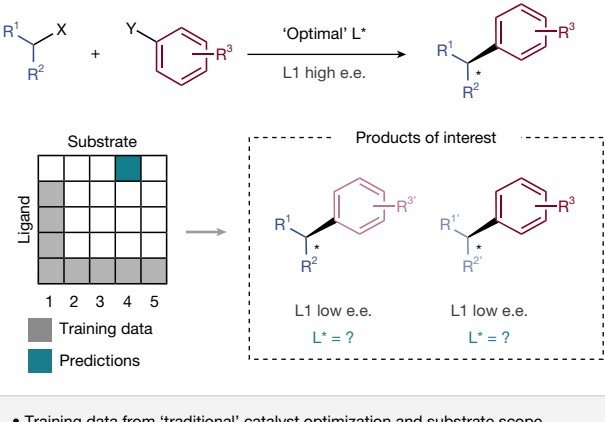

= C($sp^2$)

Ligand-controlled enantioselectivity

DFT featurization

DFT: cost-prohibitive to featurize useful number of structures

Ligand only | Ligand + substrate(s) | Representative Ni complex | Stereodetermining TSs

Increasing catalytic relevance, increasing computational cost

**c** This approach: xTB-level featurization of catalytically relevant structures for MLR

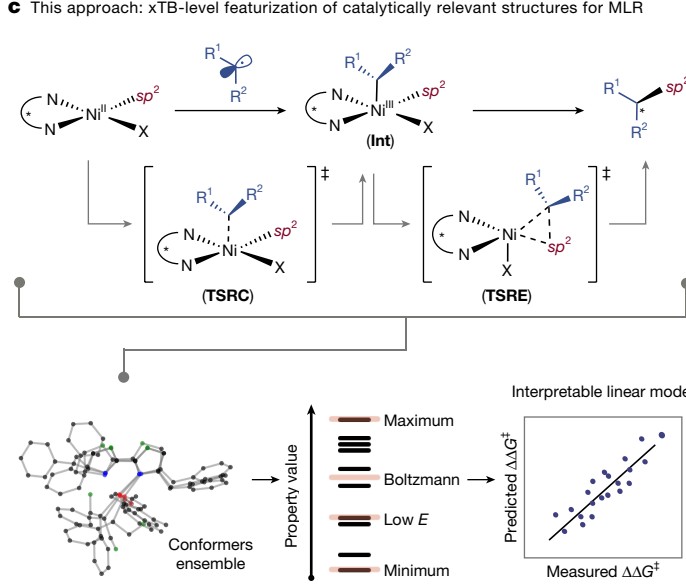

(Int) | (TSRC) | (TSRE)

Conformers ensemble

Property value: Maximum, Boltzmann, Low *E*, Minimum

Interpretable linear model

Predicted $\Delta\Delta G^{\ddagger}$ / Measured $\Delta\Delta G^{\ddagger}$

**d** Substrate-specific ligand optimization

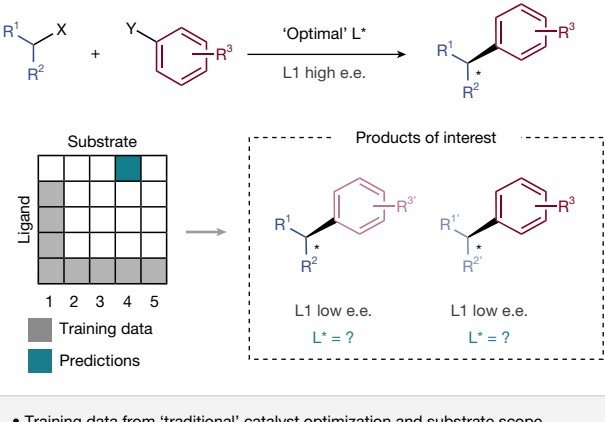

'Optimal' L*
L1 high e.e.

Substrate

Ligand

1 2 3 4 5

Training data
Predictions

Products of interest

L1 low e.e. L* = ?
L1 low e.e. L* = ?

• Training data from 'traditional' catalyst optimization and substrate scope
• 'Virtual' ligand screening for efficient e.e. optimization for product(s) of interest

**e** Transferrable models for new reaction optimization

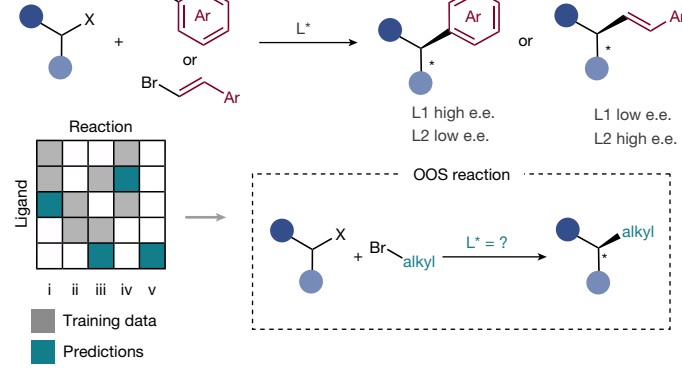

L*

L1 high e.e.
L2 low e.e.

L1 low e.e.
L2 high e.e.

Reaction

Ligand

i ii iii iv v

Training data
Predictions

OOS reaction

L* = ?

• Models trained on sparse data from mechanistically similar reactions
• Transferable to predict enantioselectivity for OOS ligands and reaction

**Fig. 1 | Development and application of transferable enantioselectivity models. a**, State-of-the-art approaches to understand and optimize asymmetric catalytic reactions, either via first-principles quantum chemical computations or the training of statistical models of enantioselectivity. **b**, Owing to high computational cost, the features used to train MLR models are typically sourced from surrogate structures to more complex catalytic species. **c**, Our proposed strategy uses low-cost features extracted from catalytically relevant structures to train MLR models applicable to multiple Ni-catalysed C($sp^3$)

couplings involving distinct electrophilic partners, enabling quantitative transfer of information to unseen ligand and reaction types. **d**, Case study 1: training MLR models on 'model system' optimization and substrate scope data enables substrate-specific ligand optimization for products reported in low e.e. **e**, Case study 2: training MLR models on e.e. data from multiple reactions sharing common elementary steps enables enantioselectivity predictions for OOS ligands and reactions. **TSRC**, radical capture transition state; **Int**, intermediate; **TSRE**, reductive elimination transition state.

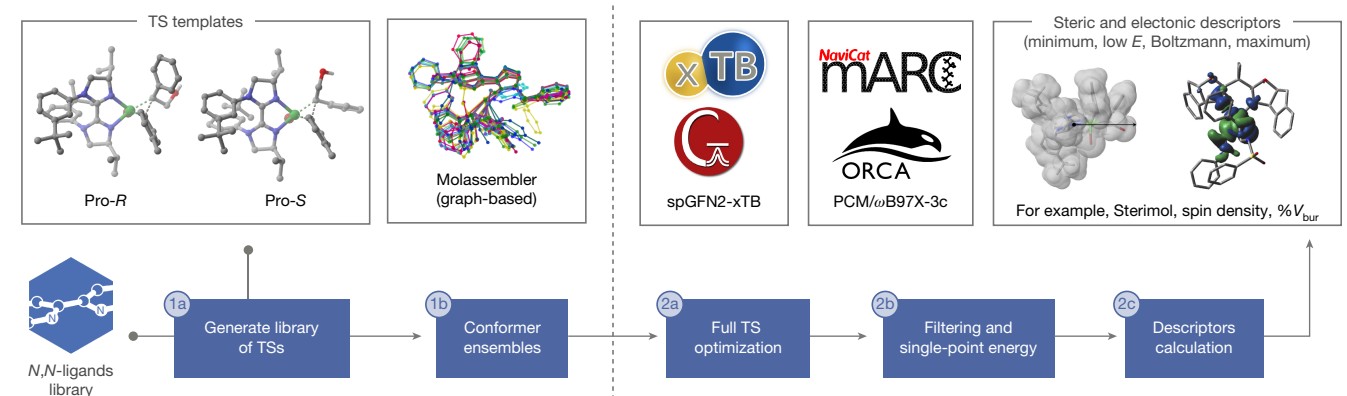

**Fig. 2 | Computational workflow.** Computational pipeline for high-throughput optimization and descriptors calculation of catalytically relevant species. (1) Automated structure generation. (2) High-throughput quantum chemical computations (PCM/$\omega$B97X-3c//spGFN2-xTB).

as a powerful method for asymmetric C–C bond formation[8,20], have relied on the identification of suitable *N,N*-ligands. Because distinct substrates have been found to necessitate different ligand classes for optimal performance, significant time and effort is dedicated to re-optimizing the reaction conditions for new coupling partners. Many of the mechanistic details of this reaction class remain under study[21]. As the substrates and ligands change, the mechanism likely undergoes subtle changes, for example, the nature of the stereodetermining step(s) may vary. In this context, a descriptor generation strategy that can characterize interactions between substrates and ligands is highly desirable for its potential to streamline enantioselectivity predictions for specific substrate combinations and new asymmetric Ni-catalysed C($sp^3$) couplings. Previous efforts have relied on experimental data collected based on a matrix of substrates and ligands[22] or via combinatorial high-throughput experimentation[23]. Conversely, our aim is to maximize knowledge extracted from sparse data where only a few combinations have been tested. This is important in the context of emerging reaction development where only limited data have been collected or published.

To this end, we report a data-driven workflow that integrates mechanistic knowledge with statistical methods by training MLR models with inexpensive physical organic parameters extracted from TSs or other catalytic intermediates (Fig. 1c). We then applied this workflow in three case studies to demonstrate its utility in streamlining reaction optimization. In the first, we showcased how sparse substrate scope data are used to train MLR models capable of predicting ligands, which promote higher enantioselectivity for a given substrate combination previously reported in lower enantiomeric excess (e.e.) (Fig. 1d). In the second, we developed a ligand prediction platform for reactions involving disparate substrates and catalysts but sharing similar elementary steps (Fig. 1e). Analysis of our mechanism-informed features suggested that reactions in different datasets (Ni-catalysed C($sp^3$)–H activation and cross-electrophile coupling (XEC)) proceed via different stereodetermining steps. As this featurization strategy captures how substrate–catalyst interactions evolve during a catalytic cycle, the resulting model was found to be transferable to a distinct reaction class featuring a ligand type previously unseen. In the third, by combining the mechanistically relevant features with our active learning platform[24], we showed how the workflow could serve as a tool for 'cold starting' a catalytic reaction development campaign, helping the synthetic chemist select a small set of promising ligands to evaluate and markedly cutting time and costs associated with exhaustive catalyst screening.

## Computational workflow and benchmarking

To address challenges in training 'holistic' MLR models that are applicable to distinct reaction classes and substrate combinations, we

developed a computational pipeline that provides access to fully optimized conformational ensembles of TSs and catalytic intermediates at an affordable computational cost. The workflow is summarized in Fig. 2, with details provided in Methods and Supplementary Information. It integrates state-of-the-art tools for automatically building, manipulating and optimizing complex three-dimensional geometries. The tools include AaronTools[25] and Molassembler[26] (steps 1a and 1b), and a wrapper script[27] that allows geometry optimizations to be performed with the spin-polarized GFN2-xTB (spGFN2-xTB) Hamiltonian[28] within the Gaussian16 software (steps 2a–2c). Compared with DFT, this reduces the cost of optimizing stationary points on a potential energy surface from months to days. Stereoelectronic parameters are extracted from the conformational ensembles and used as representation for regression methods. Despite the lower level of theory needed to ensure cost effectiveness, we envisioned that models trained with mechanistically relevant descriptors would retain the same predictive power as those using DFT-level features from simpler structures or even outperform them. Notably, previous studies have used regression models of enantioselectivity with features extracted from TS geometries[29–31]. However, optimizing these structures to a first-order saddle point at a DFT level becomes prohibitively expensive, especially when curating large datasets for virtual screening or accounting for the entire conformational space of the TSs. Alternatively, parameters have been sourced from TS-like distorted geometries where the bonds being broken or formed are constrained[32,33]. Although this approach reduces the computational cost, it limits the use of physically meaningful descriptors. Model interpretation has been further complicated by the use of nonlinear regression methods, as opposed to more interpretable linear examples[23].

To validate our featurization strategy, we revisited a Ni/photoredox-catalysed reductive coupling of styrene oxides with aryl iodides reported by our lab[14] (Fig. 3) and compared MLR models trained with features computed at different levels of theory. In our previous study, 20 bi-oxazoline (BiOx) and 9 bi-imidazole (BiIm) ligands (**L1**–**L29**) were tested on styrene oxide **1** and aryl iodide **2** as model substrates. Linear regression models of enantioselectivity (expressed as $\Delta\Delta G^\ddagger$ values; where $\Delta\Delta G^\ddagger$ is the Gibbs free energy difference between stereocontrolling TSs) were trained with features of the ligands in three coordination states: the free ligand (L*), a tetrahedral L*NiF$_2$ complex and a square planar L*Ni(*p*-tolyl)Cl complex. Geometry optimizations were carried out with the M06-2X functional and Def2-TZVP basis set: despite its cost being higher than GFN2-xTB, this level is regarded among the state-of-the-art DFT methods. Feature selection was performed via repeated, stratified nested *k*-fold cross-validation; model performance across different coordination states was further evaluated via a 5 × 2 cross-validation test (see Supplementary Information section 2.4 for full details). Our previous study showed that although more precise

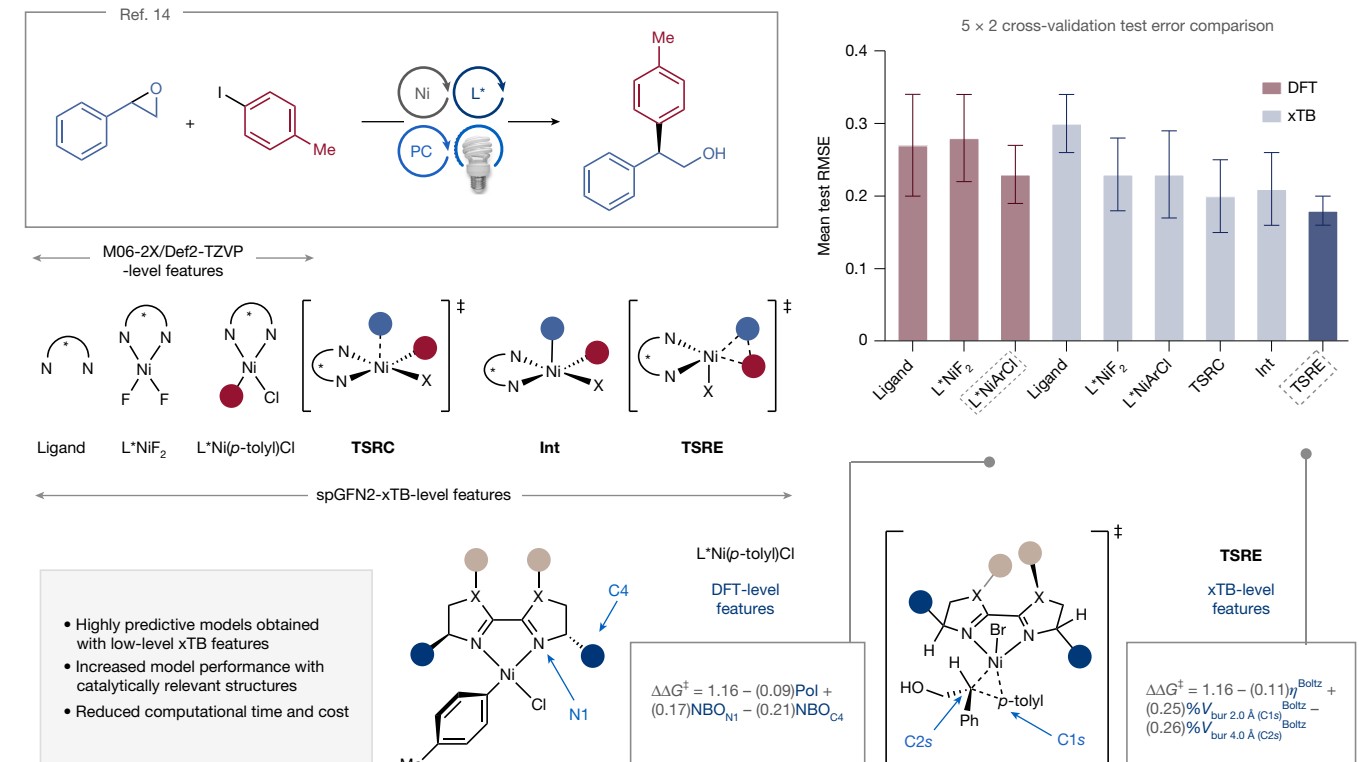

**Fig. 3 | Benchmarking.** Ni/photoredox-catalysed coupling of styrene oxides with aryl iodides[14] and evaluation of MLR models via 5 × 2 cross-validation trained with DFT-level features of surrogate structures (free ligand, L*NiF$_2$ or L*Ni(Ar)Cl) or spGFN2-xTB-level features of catalytically relevant species (**TSRC**, **Int** or **TSRE**) versus surrogate structures. The error bars in the category plot denote the standard deviation of the 5 × 2 cross-validation test. The dashed boxes denote the structure affording the best-performing model with DFT-level or spGFN2-xTB-level features. PC, photocatalyst; Pol, polarizability.

descriptors are obtained from L*Ni(*p*-tolyl)Cl, free-ligand features are sufficient in constructing a descriptive model, albeit with limited predictive accuracy (average adjusted $R^2 = 0.66$).

Using our developed featurization workflow, we acquired spGFN2-xTB-level parameters of the three coordination geometries, as well as descriptors of the stationary points presumably involved in the stereodetermining steps: radical capture (formally, oxidative ligation[8], **TSRC**) by a L*Ni(II)(C-*sp*$^2$)X species yielding a L*Ni(III)(C-*sp*$^2$)(C-*sp*$^3$)X intermediate (**Int**) and reductive elimination (**TSRE**) to form the cross-coupled product. Using the same repeated, nested *k*-fold cross-validation scheme[14], highly predictive models were obtained with the low-level features (Fig. 3), suggesting that the time and cost associated with expensive DFT computations may be saved. Furthermore, featurizing more catalytically relevant structures led to models that outperform those built with less relevant features. Hypotheses tests such as analysis of variance (Supplementary Information section 2.4) showed that the mean test root mean square error (RMSE) across the different structures and levels of theory are not equal, and that the model trained with features extracted from **TSRE** (average adjusted $R^2 = 0.83$, mean test RMSE = $0.18 \pm 0.02$ kcal mol$^{-1}$) is an improvement over the free ligand and L*NiF$_2$ features model (at the M06-2X/Def2-TZVP level). A significant difference between the performance of the **TSRC** and **TSRE** models ($P = 0.011$) was also observed. As the **TSRE** was found to be stereodetermining in our previous computational analysis of the potential energy surface for this reaction[14], using mechanistically relevant descriptors may be a strategy to indirectly probe the nature of the enantiodetermining step. The **TSRE** model contains three parameters (Fig. 3): the global hardness $\eta$ and the percent buried volume %$V_{bur}$ about the carbon atoms of the two electrophiles (labelled C1*s* for the *sp*$^2$-coupling partner within a 2.0 Å sphere, and C2*s* for the *sp*$^3$ electrophile within a 4.0 Å sphere). Because the descriptors are normalized, the relative sign and magnitude in the linear equation are indicative

of each feature's importance. Enantioselectivity is negatively correlated to hardness, calculated as the difference between the frontier molecular orbital energies. This parameter can be thought of as the resistance of a molecule in undergoing electron transfer and may suggest that faster reductive elimination is associated with a larger degree of stereocontrol, or that more electron-donating ligands afford higher e.e. values. The %$V_{bur}$ terms have similar contributions to determining enantioselectivity, albeit with opposite signs, suggesting that the ligand interactions with the aryl and benzyl groups have distinct effects.

## Case study 1

Optimizing reaction conditions, specifically the identity of the chiral ligand for asymmetric induction, is generally accomplished on model substrates; once conditions have been optimized, the reaction scope is examined. As such, enantioselectivity data for each product in a reaction scope are typically limited to the experimental results of evaluating one ligand per distinct substrate combination (Fig. 4a). This specificity-oriented approach may result in catalysts that are tailored towards one substrate type, but exhibit diminished selectivity for more diverse scope examples, limiting broad applicability of the synthetic method[34]. However, it is both time- and resource-intensive to experimentally re-evaluate a large ligand set to identify a better performer for an entire substrate scope, or even for a single substrate combination of interest[35]. Chemists also often find it difficult to intuit what catalyst modifications to make to improve enantioselectivity for accessing a specific product, or whether modifying a catalyst will lead to increased enantioselectivity at all. Furthermore, recognizing whether structural features of certain substrates prevent them from achieving high enantioselectivity, even with the optimized ligand, is challenging. This problem is further exacerbated for mechanistically complex reactions, such as Ni-catalysed XECs. As our mechanistically

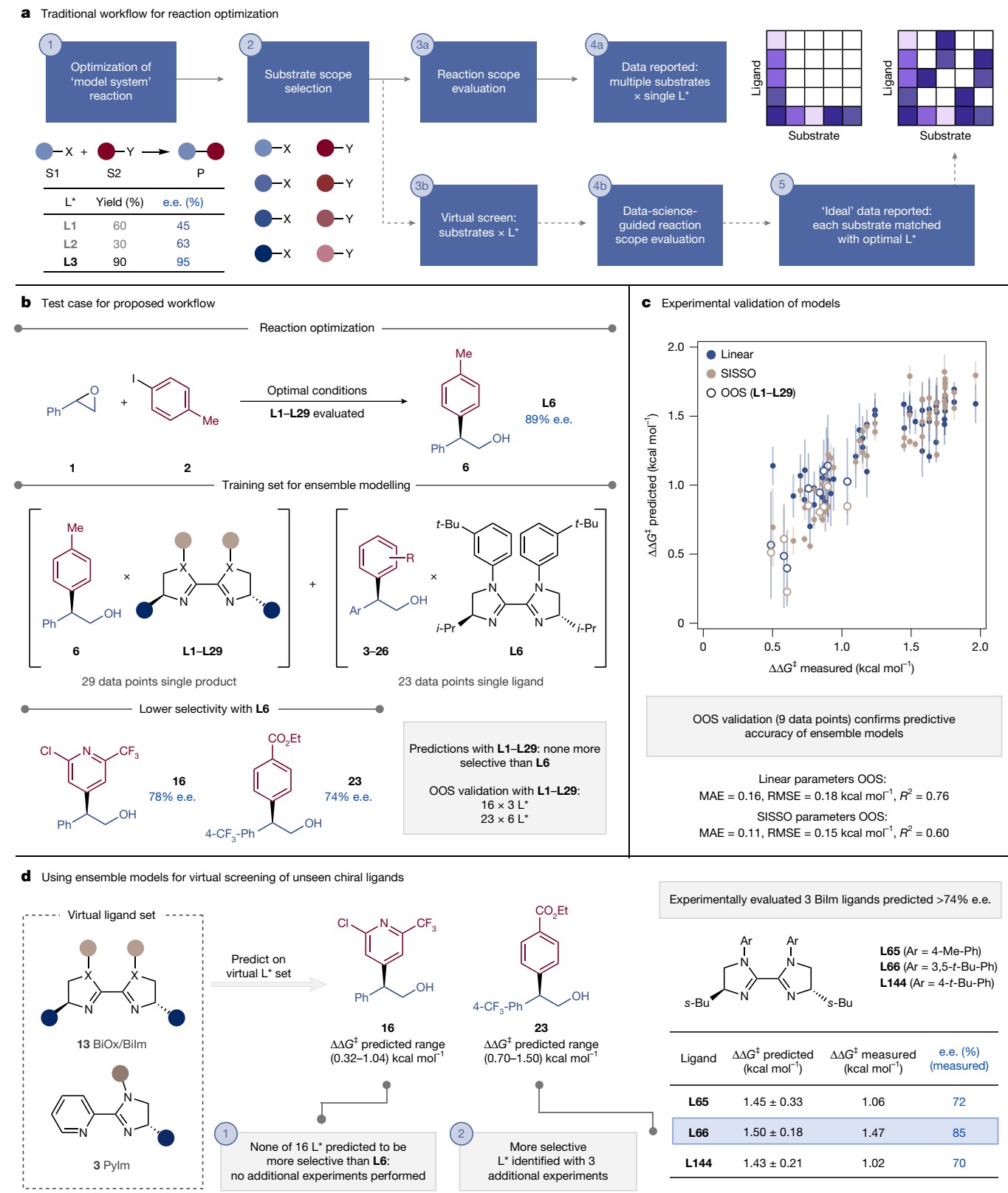

**Fig. 4 | Case study 1. a**, Traditional approach to substrate scope exploration (one catalyst screened against multiple combinations of substrates) versus catalysts × substrates virtual screening and identification of optimal ligand–substrates combinations. S1, substrate 1; S2, substrate 2; P, product. **b**, Training data and products originally reported in lower enantioselectivity selected for optimization. **c**, Ensembles of MLR models trained with linear **TSRC**–**Int**–**TSRE**

descriptors (blue) or SISSO combinations of descriptors (beige) and OOS predictions on the original **L1**–**L29** set (white circles, nine datapoints). The error bars in the parity plot denote the standard deviation of the ensemble model predictions. One poorly soluble ligand was excluded from the OOS set (Supplementary Table 5). **d**, Expanded 'virtual' set of 16 unseen ligands and experimental validation of promising candidates.

relevant features explicitly describe catalyst–substrate interactions, we proposed that we could train regression models with sparse optimization (that is, fixed model substrate, varying ligand) and scope data (that is, multiple substrates with a fixed ligand). We could then use these models to predict the enantioselectivity of scope examples with untested ligands, thereby streamlining reaction optimization for poorer-performing reaction partners. Such a prediction platform would allow us to probe synergistic interactions between catalysts and substrates while simultaneously reducing the time and cost associated with exhaustive experimental evaluation of an entire ligand–substrate matrix[22].

To achieve this, we re-examined the scope of the reductive coupling of styrene oxides and aryl iodides (Fig. 4b): 24 entries were reported in mid to high enantioselectivity with 3-*t*-Bu-*i*-Pr-BiIm ligand **L6**. We optimized and featurized **TSRC**, **Int** and **TSRE** for all the substrate combinations with **L6** and used this data, together with the **L1**–**L29** ligand screen on the model substrates, to train a MLR model. For each datapoint, a reaction representation was constructed by concatenating the **TSRC**, **Int** and **TSRE** features; using only **TSRE** parameters led to inaccurate out-of-sample (OOS) predictions, potentially owing to changes in the enantiodetermining step with varied substrates (see Supplementary Information section 3.4 for full details). To limit the number of input features, only Boltzmann-weighted average values were included for electronic descriptors, whereas minimum, maximum and average values were used for steric parameters. This 1,251 feature set was reduced with the Boruta algorithm[36], followed by selection of the most predictive combinations of 4 descriptors via repeated, stratified, nested *k*-fold cross-validation. On the basis of the 5 × 2 cross-validation test, 5 multivariate linear models were identified (each with mean test RMSE in the range 0.29–0.32 ± 0.03–0.06 kcal mol$^{-1}$), and ensemble predictions calculated as an average of these 5 models, weighted by the mean test RMSE in the 5 × 2 cross-validation scheme (see Supplementary Information section 3.5 for full details). Ensemble modelling provides a measure of uncertainty via the standard deviation of the predicted values, helping de-risk the selection of ligands to screen for reaction optimization, and results in higher predictive accuracy (ensemble full data-fit MAE = 0.17 kcal mol$^{-1}$, RMSE = 0.21 kcal mol$^{-1}$, $R^2$ = 0.75; Fig. 4c; MAE is the mean absolute error).

We next sought to experimentally evaluate whether this mechanistically informed modelling strategy would be effective at predicting a catalyst specifically tailored to a particular substrate combination. Among the 24 cross-coupled products surveyed[14], **16** and **23** had been reported in lower enantioselectivity (78% and 74%, respectively, versus 89% e.e. for **3**; Fig. 4b) and were selected for virtual screening; however, we found that none of the original **L1**–**L29** ligands were predicted to be an improvement over **L6**. Similarly, we virtually screened ligands **L1**–**L29** against other substrate combinations (specifically, products **15**, **17**, **22** and **26** from ref. 14), but we found that this ligand set was not expected to afford e.e. values higher than those obtained with **L6**. Before making predictions on additional ligands from a virtual library, we sought to experimentally validate the ensemble model. We tested 4 BiOx and 5 BiIm ligands in the cross-coupling to form **16** or **23** and, as expected, this resulted in low to mid e.e. values (39–71% e.e., true negatives; Supplementary Tables 4 and 5), confirming the model's good OOS predictive accuracy (MAE = 0.16 kcal mol$^{-1}$, RMSE = 0.18 kcal mol$^{-1}$, $R^2$ = 0.76; Fig. 4c). Because these OOS predictions were associated with a high degree of uncertainty (±0.20–0.39 kcal mol$^{-1}$), we turned to the sure independence screening and sparsifying operator (SISSO) algorithm[37] to introduce nonlinear descriptor relationships into modelling. We have previously found this strategy to improve test accuracy by describing highly complex and dynamic catalyst–substrate interactions[6]. A high-performing ensemble model of SISSO-generated descriptors was trained (MAE = 0.14 kcal mol$^{-1}$, RMSE = 0.17 kcal mol$^{-1}$, $R^2$ = 0.82; see Supplementary Information section 3.6 for full details), but afforded similar OOS predictions (MAE = 0.11 kcal mol$^{-1}$,

RMSE = 0.15 kcal mol$^{-1}$, $R^2$ = 0.60; Fig. 4c), albeit with a lower degree of uncertainty (±0.04–0.23 kcal mol$^{-1}$).

Virtually screening the **L1**–**L29** set with nonlinear parameters similarly suggested that none of the ligands originally evaluated on the model substrates would afford **16** or **23** in improved enantioselectivity. Notably, in the absence of such models, 56 additional experiments (2 substrates × 28 ligands) would be required to arrive at this conclusion. To evaluate whether this virtual screening approach could be extended further (while maintaining experimental practicality) to identify a better-performing catalyst, we expanded our virtual search space to 16 additional BiOx, BiIm and pyridine-imidazoline (PyIm) ligands accessible in our lab (Supplementary Fig. 10). For **16**, none of the additional 16 ligands in this set were predicted to afford an increase in enantioselectivity, suggesting that **L6** is optimal in the surveyed ligand space. For cross-coupled product **23**, both ensemble models predicted **L65**, **L66** and **L144** to be more selective than **L6** and were chosen for experimental validation (Fig. 4d; see Supplementary Fig. 7 for additional ligands tested). Excitingly, **L66** afforded **23** with greater enantioselectivity (85% versus 74% e.e. using **L6**). The predicted ΔΔ$G^{\ddagger}$ values of **L65** and **L144** were higher than the measured ones; in addition, the 11% increase in e.e. achieved for product **23** (that is, from a ΔΔ$G^{\ddagger}$ value of 1.13 to 1.47 kcal mol$^{-1}$) is modest, consistent with this reaction having been extensively optimized in previous work[14,24]. Nevertheless, this prediction corresponded to an extrapolation equal to twice the OOS MAE of the ensemble MLR model with linear parameters (Fig. 4c). Moreover, when screening the virtual set (Supplementary Fig. 10), inferences were made on combinations of entirely unseen ligands and products that have been seen only once in the training set. Therefore, the identification of **L66** as a (marginally) superior ligand for product **23** suggests that our low-cost featurization workflow can be a useful tool in streamlining reaction optimization for challenging substrates.

## Case study 2

We next questioned whether we could use our featurization workflow to develop comprehensive models encompassing multiple and diverse reaction profiles to transfer this machine-learned information and provide enantioselectivity predictions on OOS ligands and reactions. Sourcing descriptors from catalytically relevant structures (that is, **TSRC**, **Int** and **TSRE**) allows us to simultaneously account for the identity of the catalyst and coupling partners. This obviates the need to parameterize each reaction component separately, thereby reducing the risk of overfitting on sparse datasets. To this end, we curated two datasets of Ni-catalysed C($sp^3$) couplings: the first (reactions A–C; Fig. 5a) comprises the enantioselective benzylic C($sp^3$)–H bond arylation, acylation and alkenylation via dual photoredox/nickel catalysis[38–40]. Given the similar reaction conditions and the presence of a common coupling partner, we reasoned that it would be an ideal proof-of-principle case study to test our hypothesis. Modelling the second dataset (reactions D–G; Fig. 5a), which comprises the asymmetric reductive arylation and alkenylation of *N*-sulfonyl styrenyl aziridines or benzylic chlorides[41–44], presents a greater challenge as both components are varied simultaneously and different reaction conditions are used (for example, electrochemical reduction versus stoichiometric Mn$^0$ reductant). We also selected the coupling of styrenyl aziridines and primary alkyl bromides[45] (reaction H; Fig. 5a) to test the model's ability to make effective OOS predictions on an unseen reaction class (C($sp^3$)–C($sp^3$) XEC).

Ensembles of 4-feature MLR models were trained using the **TSRC**–**Int**–**TSRE** reaction representation as input, reduced with the Boruta algorithm, with combinations of descriptors selected via the repeated, stratified, nested *k*-fold and 5 × 2 cross-validation schemes (see Supplementary Information sections 4.1 and 4.6 for full details). For the C($sp^3$)–H bond activation dataset (reactions A–C; Fig. 5a), excellent correlations between the predicted and measured enantioselectivities

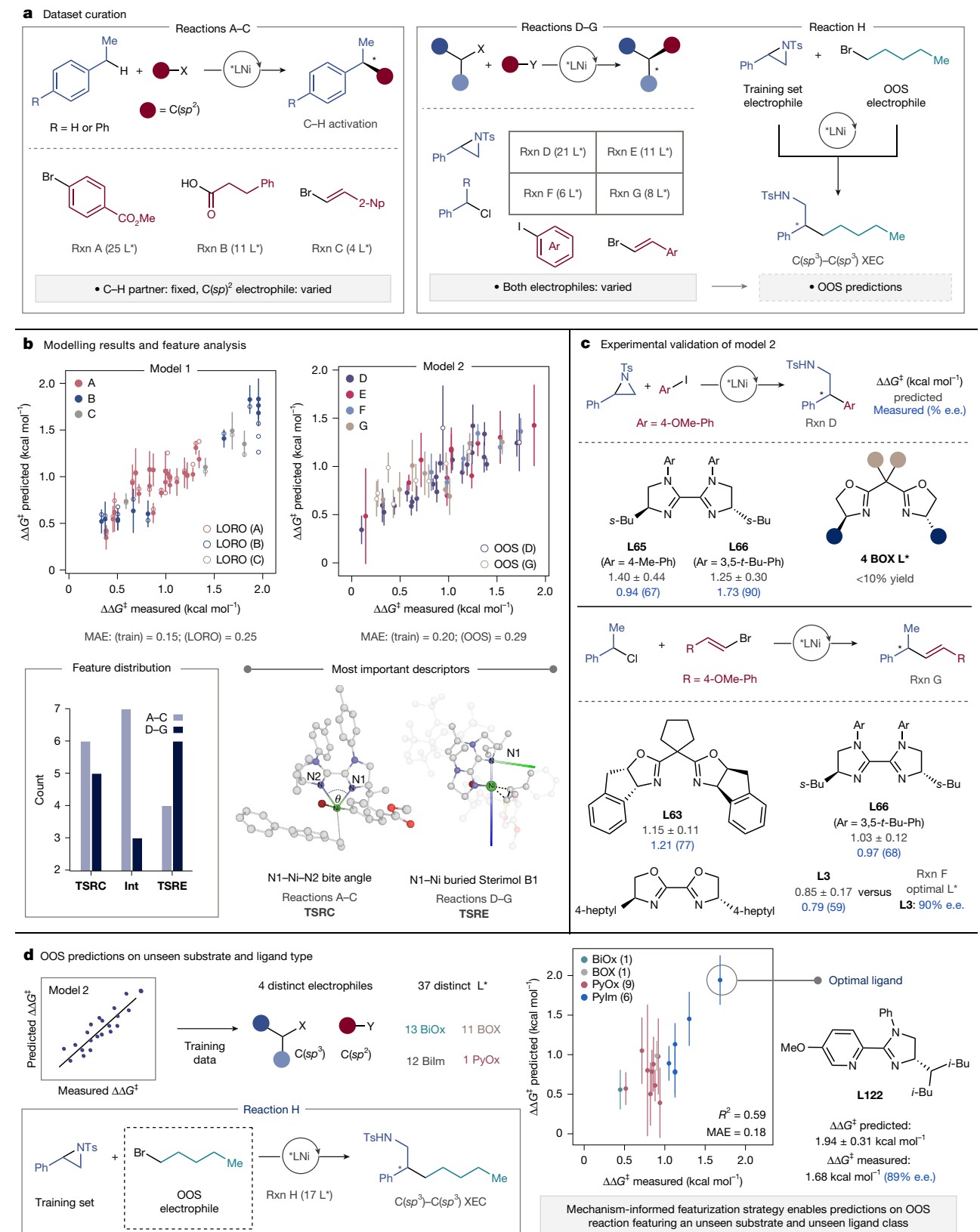

**Fig. 5 | Case study 2. a,** Dataset design for case study 2. Reactions A–C, C–H functionalization reactions reported with similar conditions and one fixed coupling partner. Reactions D–G, C($sp^3$)–C($sp^2$) XECs reported with different conditions, varying both coupling partners. Reaction H, C($sp^3$)–C($sp^3$) XEC used as an OOS test case for MLR model 2 trained on data from reactions D–G. **b,** Ensemble modelling results and analysis of feature importance. The blue axis indicates the Sterimol *L* vector, while the green one indicates the B1 vector.

Shaded atoms are excluded from the 4.5 Å sphere. **c,** Experimental validation of model 2. OOS ligands were screened in reactions D and G, according to reported reaction procedures[44,49]. Ligands affording less than one catalyst turnover were excluded from the OOS set (Supplementary Table 16). **d,** Evaluation of predictive ability of model 2 on reaction H featuring an unseen electrophile and ligand class. The error bars in the parity plots denote the standard deviation of the ensemble model predictions.

were established (ensemble full data-fit MAE = 0.15 kcal mol⁻¹, RMSE = 0.18 kcal mol⁻¹, $R^2$ = 0.88; Fig. 5b). To test the model's capability of matching patterns across distinct reaction types, we performed 'leave-one-reaction-out' (LORO) analysis, whereby each transformation is varied as a validation set and the ensemble model retrained on the other two. Through this statistical evaluation, good predictive accuracy was found (average MAE = 0.25 ± 0.09 kcal mol⁻¹, $R^2$ = 0.79 ± 0.11). Even reaction C could be predicted with diminished yet acceptable accuracy (MAE = 0.30 kcal mol⁻¹) despite the limited number of datapoints (4 out of 38 reactions) and the sparse nature of the entire dataset, suggesting that the model captured specific structural changes even if a substrate class is not adequately expressed in the training set.

Encouraged by these results, we sought to extend this approach to training a comprehensive model capable of capturing interactions between four varying electrophiles (reactions D–G; Fig. 5a). The resulting ensemble model showed adequate statistics (MAE = 0.20 kcal mol⁻¹, RMSE = 0.23 kcal mol⁻¹, $R^2$ = 0.74; Fig. 5b), even though LORO analysis suggested that reactions involving benzylic chlorides and bisoxazoline (BOX) ligands (that is, reactions F and G) were more difficult to learn (MAE = 0.24 ± 0.05 kcal mol⁻¹, $R^2$ = 0.53 ± 0.27; Supplementary Table 19). This is probably because reactions F and G were reported only with BiOx and BOX ligands, respectively, whereas a wider range of ligand types (BiIm, BiOx or BOX) were tested in reactions D and E. As a result, some predictions are associated with a large degree of uncertainty, for example, only one 2-(2-pyridyl)oxazoline (PyOx) ligand was tested (in reaction E) leading to ±0.50 kcal mol⁻¹ in its predicted $\Delta\Delta G^{\ddagger}$ value. To further probe the predictive ability of this model, we experimentally screened ten distinct ligands in reactions D and G, including classes that had not previously been reported with a specific combination of electrophiles. Specifically, we tested four BOX (unseen) and two BiIm ligands in the coupling of styrenyl aziridines and aryl iodides (reaction D; Supplementary Table 16), as well as four BiOx and two BiIm (unseen) and two BOX ligands in the coupling of benzylic chlorides and vinyl bromides (reaction G; Supplementary Table 17). Although catalysts featuring BOX ligands resulted in low or irreproducible yields for reaction D (less than one turnover; see Supplementary Information section 4.4 for details), modest yields and enantioselectivities were measured for BiOx and BiIm ligands in reaction G. Overall, these OOS predictions (MAE = 0.29 kcal mol⁻¹, RMSE = 0.36 kcal mol⁻¹, $R^2$ = 0.59) highlight the model's ability to describe catalyst–substrate interactions even for ligand classes that were previously unseen. This is particularly valuable for Ni-catalysed XECs, where changing one electrophile may result in the need for a completely different ligand type for optimal performance. For example, 4-heptyl-BiOx ligand **L3**, which was developed to catalyse the coupling of (hetero)aryl iodides and benzylic chlorides (reaction F, 90% e.e.)[43], performs poorly in reaction G (predicted $\Delta\Delta G^{\ddagger}$ = 0.85 ± 0.17 kcal mol⁻¹, measured = 0.79 kcal mol⁻¹, 59% e.e.), where a heteroaryl iodide is replaced with a vinyl bromide. This modelling approach may therefore be useful to identify promising ligand types for new coupling partners. The larger uncertainties associated with predictions for BiIm ligands in reaction D could be attributed to changes in conditions compared with those of ref. 41. Of note, we chose to model XECs carried out in different solvents (dimethylacetamide, 1,4-dioxane and tetrahydrofuran) and not include solvent parameters in the MLR analysis[1]. Although this may have resulted in reduced correlations, our results suggest that featurizing only the structures involved in the putative stereodetermining steps is adequate to build predictive models across distinct reaction types.

To gain further insight into the two regression models, we calculated the frequency with which features from distinct stationary points appear in the ensembles. Interestingly, the distribution across the **TSRC**–**Int**–**TSRE** structures is significantly different for reactions A–C and reactions D–G (Fig. 5b). For the C($sp^3$)–H activation dataset, the most predictive features are predominantly sourced from the first step of the C($sp^2$)–C($sp^3$) bond-forming mechanism (that is, **TSRC** and **Int**).

The feature that appears most frequently in the ensemble (8 out of 10 MLR models) is the N1–Ni–N2 angle of the radical capture TS (Fig. 5b). For the second dataset, the features in the ensemble are predominantly sourced from either **TSRE** or **TSRC**, with **Int** descriptors appearing less frequently. The most important descriptor (8 out of 10 MLR models) is the **TSRE** N1–Ni B1 Sterimol parameter, that is, the minimum width measured perpendicular to the N–Ni bond[13] (within a 4.5 Å sphere; Fig. 5b). This feature describes the first coordination sphere of the Ni atom and accounts for the catalyst–substrate interaction effects. Notably, the ability to generate descriptors from catalytic intermediates and TSs and analyse features importance for two distinct datasets provided insight into mechanistic differences between the two reaction classes examined – specifically, a likely change in the enantiodetermining step.

As a final assessment of the model's domain of applicability, we evaluated its ability to transfer the structural features identified as important for enantioselective catalysis to entirely unseen ligand and reaction classes. The alkylative aziridine ring opening[45] (reaction H; Fig. 5c) is a stereoconvergent C($sp^3$)–C($sp^3$) XEC reported to proceed in optimal yield and e.e. with PyIm ligands, a class that is not contained in the training dataset (reactions D–G). Furthermore, 9 out of 17 of the ligands screened were PyOx, which is a class included in only 1 out of 46 reactions seen by the trained model. Despite these significant differences, the presence of a known coupling partner (N-tosyl styrenyl aziridine) prompted us to test the OOS predictive accuracy of the ensemble model on this dataset, which was found to be high (MAE = 0.18, RMSE = 0.24 kcal mol⁻¹, $R^2$ = 0.59, Fig. 5c). The model's ability to transfer information learned on sparse C($sp^2$)–C($sp^3$) coupling data to a new XEC reaction is exciting, especially considering the mechanistic complexity of this C($sp^3$)–C($sp^3$) cross-coupling. These results suggest that this type of modelling approach should enable data-science-guided optimization of asymmetric transformations that share mechanistic features with previously published transformations.

## Case study 3

Having validated our featurization strategy on retrospective studies, we next sought to apply it to the prospective optimization of an unreported asymmetric reaction, specifically the coupling of α-oxy radicals generated from benzylic acetals with aryl iodides[46] (reaction I; Fig. 6a). We sought to (1) explore a large virtual ligand space to demonstrate the scalability of the featurization workflow, and (2) identify promising ligands in a limited number of experiments. As this C($sp^3$)-coupling partner is outside the domain of applicability of ensemble model 2 (see Supplementary Information section 5.1 for full details), we integrated the mechanism-informed featurization strategy with our open-source active learning platform[24] EDBO+ to leverage Bayesian theory for the optimization of enantioselectivity. Parameters of **TSRC**, **Int** and **TSRE** for 195 chiral N,N-ligands in reaction I were collected, which corresponded to the optimization of approximately 70,000 geometries (completed in about 2 weeks). To avoid a combinatorial explosion of search space, optimization efforts were focused on evaluating ligands for enantioselectivity and, therefore, reaction conditions (for example, solvent, reductant, temperature) were not varied (see Supplementary Information section 5.3 for experimental details). Bayesian optimization is an ideal alternative to high-throughput screening in the context of asymmetric Ni-catalysed cross-coupling reactions given the limited commercial availability of chiral ligands (only about 47% of the ligands virtually screened are commercial) and time associated with multi-step synthesis of non-commercial examples.

As demonstrated in case study 2, reactions involving distinct coupling partners may be simultaneously modelled using the mechanistically informed features. We therefore pre-trained a Gaussian process regression (GPR) model on **TSRC**–**Int**–**TSRE** descriptors of sparse data from published enantioselective Ni-catalysed C($sp^3$) cross-coupling reactions, maximizing information extracted from the literature

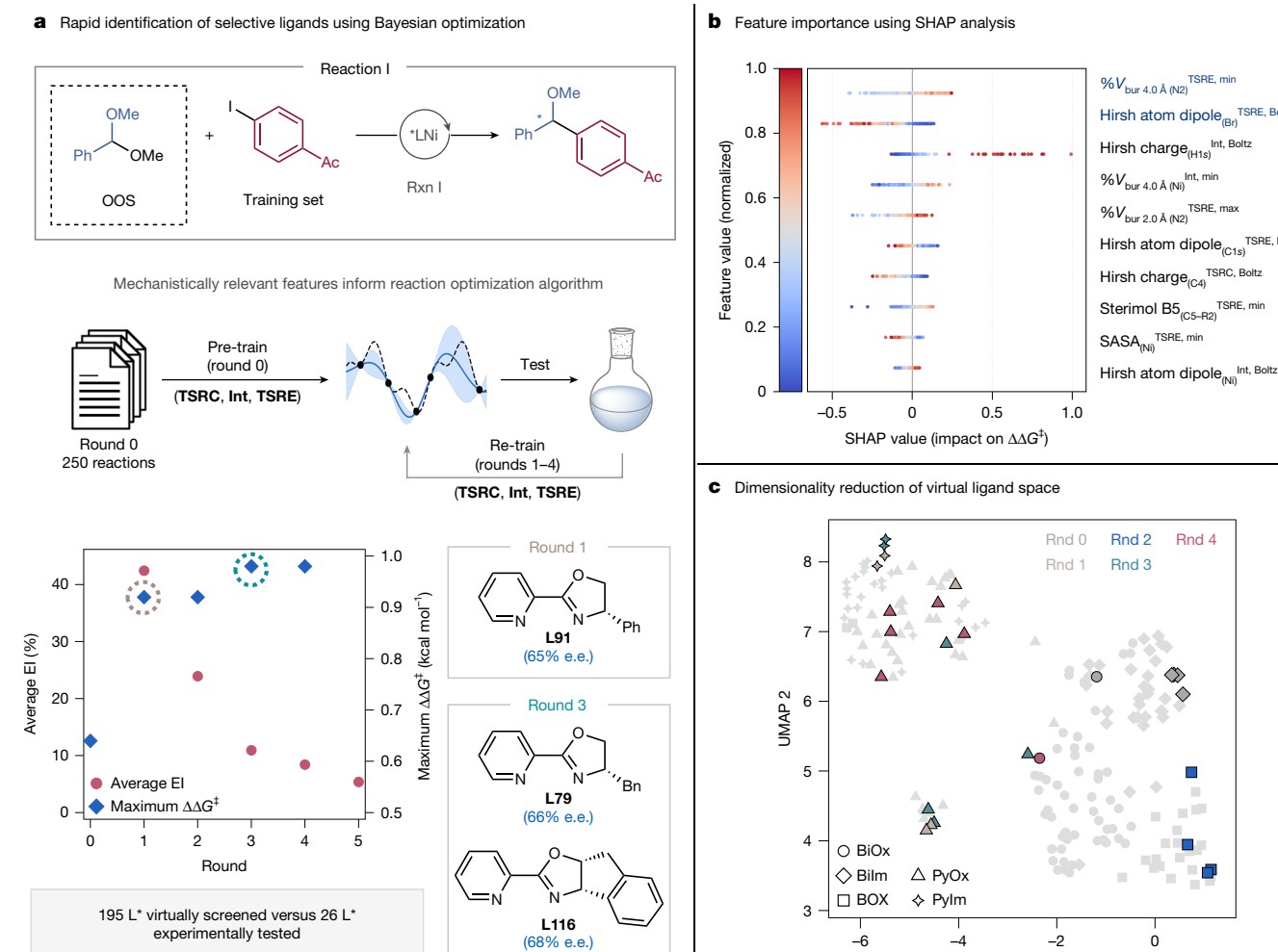

**Fig. 6 | Case study 3. a**, Prospective application of the featurization strategy: Ni-catalysed coupling of benzylic acetals and aryl iodides. The C($sp^3$)-coupling partner is unseen (that is, missing from the training data). The GPR model is trained on literature data before the start of the active learning campaign using EDBO+. In the plot, max $\Delta\Delta G^{\ddagger}$ indicates the maximum enantioselectivity achieved up to round $n$. EI, expected improvement. **b**, Beeswarm plot highlighting the ten most important features contributing to the model's regression, ordered according to their mean absolute SHAP value. **c**, UMAP chemical space representation of the ligand space for reaction I. Dimensionality reduction was performed using the top-ranked features whose mean absolute SHAP contribution to $\Delta\Delta G^{\ddagger}$ was ≥0.10 kcal mol$^{-1}$, retaining features in descending importance until they collectively accounted for ≥90% of the total mean absolute SHAP importance. SASA, solvent-accessible surface area.

(round 0; see Supplementary Information section 5.2 for full details). This dataset is available on GitHub to help 'cold start' the optimization of other transformations. At the beginning of the active learning campaign (round 0), benzylic acetals are entirely unseen coupling partners; the GPR model is progressively retrained on small batches of experiments (4–6 ligands screened per round) until the expected improvement of $\Delta\Delta G^{\ddagger}$ indicates diminishing return to performing additional experiments (average expected improvement of about 5% for round 5; Fig. 6a) and the model's prediction accuracy is acceptable (MAE = 0.33 kcal mol$^{-1}$ at round 4; Supplementary Fig. 13).

Despite the modest enantioselectivity achieved in reaction I (maximum $\Delta\Delta G^{\ddagger}$ = 0.98 kcal mol$^{-1}$, 68% e.e.), EDBO+ identified the likely optimal ligands in this search space within only 4 rounds, that is, 26 experiments. Notably, the top-performing ligand class, that is, PyOx (see ligands **L91**, **L79** and **L116** in Fig. 6a) is different from the optimal classes for the literature reactions used to pre-train the GPR model (where BiIm, BiOx, PyIm or BOX ligands are top-performing). To determine the most influential descriptors in the GPR model, we performed SHAP (SHapley Additive exPlanations) analysis[47] (Fig. 6b), which identified %$V_{bur}$ parameters and Hirshfeld charges as the most significant contributors to the model's outcome; 36% of the most important features are extracted from **TSRE**, followed by **Int** (34%) and **TSRC** (30%). We then reduced this subset of the feature space to two dimensions using the uniform manifold approximation and projection (UMAP)[48] method, revealing how distinct ligand classes are clustered together (Fig. 6c). During the optimization campaign, EDBO+ explores and exploits these different ligand types, for example, BOX in round 2 and PyOx in rounds 3 and 4, identifying **L116** (and **L79**) as optimal. By integrating the mechanism-informed featurization strategy with an existing data-science-guided reaction optimization platform, we were able to leverage literature data and rapidly identify a selective ligand class for an asymmetric cross-coupling reaction.

## Discussion

We have developed a workflow to predict the enantioselectivity of Ni-catalysed C($sp^3$) couplings where multiple reaction components are varied simultaneously using only sparse data for training. Parameterizing regression models with descriptors extracted from catalytically relevant structures increases the models' accuracy and domain of applicability. Owing to the mechanistic complexity of Ni-catalysed C($sp^3$) couplings, the input representation is large, which makes

identifying the most important descriptors challenging. We have addressed this problem via repeated, nested $k$-fold cross-validation and ensemble modelling, but less expensive approaches are still desired. The low-cost, mechanism-informed features obtained via our workflow allowed us to identify optimal catalyst–substrate combinations and make accurate predictions on OOS ligand and substrate classes. The added value of this approach is that limited data of known reactions may be used to train models applicable to unknown transformations. While we envision that it will facilitate the development of transfer-learning approaches for the discovery of synthetic methods, alternative solutions to extract detailed mechanistic information and unambiguously identify the nature of the enantiodetermining step for a given combination of ligand and substrates will further expand the scope of this workflow.

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

## Methods

### Computational workflow

Our pipeline is summarized in Fig. 2, and further details are provided in Supplementary Information and on GitHub at https://github.com/SigmanGroup/HT_TSs_Opt. In step 1, ligands or substituents from a user-defined library are mapped onto previously computed structures (for example, the diastereomeric TSs for the stereodetermining step) with AaronTools[25], enabling the automated construction of initial geometries for multiple ligands and/or substrates. To gain access to a full conformational space, Molassembler[10,26] is used to interpret the coordinates of each template into a graph, from which about 250 conformers are generated and projected back into three-dimensional coordinates, constraining the first coordination sphere of the Ni atom to ensure geometries of good quality for further optimization. Because Molassembler generates conformers via graph-based treatment of chemical structures, rather than by performing metadynamics simulations[50], step 1 is exceptionally fast: for a library of 29 BiOx and BiIm ligands (see benchmarking), about 10,000 reductive elimination pseudo-TSs are generated in just over 3 minutes.

In step 2, the resulting ensembles of geometries are refined at the GFN2-xTB level[51] with key bonds and angles (for example, $Ni-C(sp^2)$, $Ni-C(sp^3)$) being constrained, followed by optimization to either a first-order saddle point or a minimum on the potential energy surface with the spin-polarized GFN2-xTB Hamiltonian[28] and Gaussian16's (Rev. C.01)[52] Berny algorithm. This is achieved by using a wrapper script[27] that allows xtb to be used as part of Gaussian, resulting in significant time acceleration over DFT-level optimizations (Supplementary Information section 2.2). Successfully converged geometries are automatically filtered according to their vibrational frequencies (minima with zero imaginary frequencies, TSs with one imaginary frequency of the correct magnitude); the optimal number of unique structures needed to completely cover the conformational space, within an energetic window of 10 kcal mol$^{-1}$, is then selected via clustering performed with marc[53], followed by single-point energy computations at the PCM/ωB97X-3c level with the ORCA package[54]. This composite method has been shown to be on par with standard hybrid DFT methods in a quadruple-zeta basis set at a fraction of the computational cost[55]. Next, molecule-, atom- and bond-level descriptors of the conserved moiety are computed at the spGFN2-xTB level with the xTB-Gaussian wrapper, then collected and processed using our recently reported[56] Get Properties Jupyter notebook. For each TS or intermediate, 189 geometric and steric properties (for example, bond lengths, angles and dihedrals, percent buried volumes, Sterimol values, and so on) and 75 electronic descriptors (for example, frontier molecular orbital energies, Hirshfeld charges, spin densities, dispersion potentials and so on) are computed. To describe the dynamic range of conformers a structure can adopt (within a 5 kcal mol$^{-1}$ energetic window based on the Gibbs free energies calculated as the sum of the ωB97X-3c energies and the spGFN2-xTB thermal corrections), the minimum, maximum, Boltzmann-weighted average and the property value from the lowest-energy conformer are collected for each feature, yielding a total of 756 parameters per structure. This ensures that conformational flexibility is well described, which is essential to establish accurate structure–selectivity relationships.

### Data availability

Procedures for the experimental validation of the MLR models, OOS ligand screening data, synthesis of the substrates and ligands, and characterization data for products **16**, **23**, reactions D and G, and for ligands **L16**, **L28**–**29** and **L144** are available in the Supplementary Information. Tabulated enantioselectivity data (measured and predicted), details of the features used in the MLR and GPR models, and Excel spreadsheets with input features are also included in the Supplementary Information and on GitHub at https://github.com/SigmanGroup/HT_TSs_Opt, along with XYZ files of the templates used in the computational workflow. Excel spreadsheets and the scripts comprising the computational workflow are also available via Zenodo at https://doi.org/10.5281/zenodo.18356432 (ref. 57).

### Code availability

The computational workflow comprises multiple scripts written in Python and Bash and two Jupyter notebooks. These are available on GitHub at https://github.com/SigmanGroup/HT_TSs_Opt and via Zenodo at https://doi.org/10.5281/zenodo.18356432 (ref. 57).

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

**Acknowledgements** S.G. acknowledges funding from the Swiss National Science Foundation (SNSF, Postdoc.Mobility, grant number 222115) and support and resources from the Center for High Performance Computing (CHPC) at the University of Utah. S.G., E.M.B., A.G.D. and M.S.S. acknowledge financial support through the NSF Center for Computer Assisted Synthesis (C-CAS) under grant CHE-2202693. These studies were supported by shared instrumentation grants from the National Science Foundation under equipment grant number CHE-1048804 and the NIH Office of Research Infrastructure Programs under grant number S10OD028644. We acknowledge A. LeSueur for his contributions to the statistical modelling scripts of the Sigman Lab, which supported this work; T. Wild, W. Williams, J. Schleinitz and S. E. Reisman for their curation of a *N,N*-ligands library, which inspired and helped initiate this work; and M. A. Borden for her work on asymmetric Ni-catalysed cross-coupling of benzylic acetals.

**Author contributions** S.G. conceived of the project with help from E.M.B., A.G.D. and M.S.S. S.G. developed the computational workflow, curated the data, performed the computations and trained the statistical models. E.M.B. carried out all the experimental work and helped analyse the results. S.G. and E.M.B. wrote the paper with help and feedback from M.S.S. and A.G.D., who provided supervision throughout.

**Competing interests** The authors declare no competing interests.

**Additional information**
**Correspondence and requests for materials** should be addressed to Abigail G. Doyle or Matthew S. Sigman.
