## [Peer Review File · Nature]

Transferable enantioselectivity models from sparse data

Corresponding Author: Professor Abigail Doyle

Version 0:

Reviewer comments:

Referee #1

(Remarks to the Author)

The manuscript "Low-Cost Mechanism-Informed Features Enable Transferable Enantioselectivity Predictions from Sparse Data" is a natural evolution of the field of machine learning methods for synthetic chemistry. It borrows ideas of carefully crafted mechanistic descriptors of reactions, for example the systematic studies by Carles Bo (e.g., 10.1021/ci700181v) and of mechanism-linked relationships that could guide ML-based workflows, e.g., by Mikhail Kabeshov. The workflow for generation of DFT-level descriptors that are mechanistically relevant is also a natural extension of where the field is going. One recent example is the work from AstraZeneca computational group on automated DFT pipeline Maize for generating inputs for ML. Thus, this paper reads as a welcome manifestation of ideas that were maturing for a while in the field: bringing mechanistic knowledge to ML, automating identification of relevant mechanistic information and ensuring the extrapolation ability of the models through figuring out the key mechanistic features that feed data into ML models. My recommendation would be to try to stress-test the approach and clearly identify the bounds of when it breaks. There are way too many papers in the field of ML for chemistry that over-promise and are not scrutinising themselves enough.

(Remarks on code availability)

Referee #2

(Remarks to the Author)

The work from Doyle and Sigman presents a novel computational workflow for predicting enantioselectivity in nickel-catalyzed asymmetric cross-coupling reactions using low-cost, mechanism-informed features. The method involves generating descriptors from transition states (TSs) and catalytic intermediates optimized with the spin-polarized GFN2-xTB Hamiltonian, significantly reducing computational costs compared to traditional DFT methods. These descriptors are used to train multivariate linear regression (MLR) models, enabling accurate predictions of enantioselectivity even with sparse data. The approach is validated through case studies, demonstrating its ability to optimize ligands for poorly performing substrates and predict outcomes for unseen reactions and ligand classes. Thus, I support the publication of this work in Nature after addressing the following issues:

1. While the Boruta algorithm and cross-validation are effective, incorporating more advanced feature selection techniques (e.g., SHAP analysis or graph-based descriptors) could further improve model interpretability and accuracy.
2. Expanding beyond linear regression to include nonlinear methods (e.g., neural networks or kernel-based models) may better capture complex catalyst-substrate interactions, especially for highly flexible systems.
3. The workflow assumes specific TSs (e.g., radical capture or reductive elimination) as enantio-determining step. If the authors could include alternative mechanistic pathways (e.g., concerted mechanisms) could enhance robustness, particularly for controversial reactions.
4. The authors are suggested to couple the computational predictions with automated high-throughput experimentation could accelerate iterative optimization and provide rapid feedback for model refinement.

(Remarks on code availability)

Referee #3

(Remarks to the Author)

I co-reviewed this manuscript with one of the reviewers who provided the listed reports.

(Remarks on code availability)

On the code, some of the .xlsx files have different labels on the first row as what it seems to be expected from the code. Therefore, I would suggest reviewing the code provided on GitHub runs with the same files which are present in the same folder. An example run would also be helpful to include in the README.md file details on how others could run the code for one of the benchmarks, with the specific flags and file names. One example present is the inconsistency in the Ligand Class label keys that change from case study 1 to case study 2 ("Class" in "Case_Study_1_Linear_Random.xlsx", to "Lig_Class" in "Case_Study_2_CH_Functionalization.xlsx").

Referee #4

(Remarks to the Author)

The manuscript presents a workflow to create multivariate linear regression models based on a semi-empirical quantum approach (rather than standard DFT) with the objective of predicting the performance of unseen reactions. The use of semi-empirical quantum approaches, instead of the nowadays standard DFT calculations, sacrifices performance for productivity and, consequently, the authors argue that this approach should be considered as a "first-line triage" tool for catalyst discovery. The idea of using low-cost, low-level predictors for efficient massive virtual screenings is not new, but the combination with MLR models could be interesting. However, I find the manuscript insufficient to support publication in Nature, mainly based on the following weaknesses:

1. Scale of demonstration.

A central selling point of the manuscript is scalability at low computational cost. Yet, in case study 1 they only evaluated 16 new ligands, which undermines their claim. If the methodology is indeed low-cost and scalable, one would expect to see hundreds of ligands virtually screened to showcase its full potential, or even an application to a novel reaction class beyond those already published by the same group. The current scope (two Ni-catalyzed case studies, both related to previously published work by the authors in JACS) feels insufficient to demonstrate broad applicability.

2. Benchmarking against conventional approaches.

The efficiency of the method is framed by comparing "3 experiments (vs. 45 virtually screened)." However, no direct benchmark is provided against brute-force or high-throughput experimental screening, which would produce a larger number of results and, more importantly, deliver definitive outcomes without uncertainty. A quantitative comparison of time, cost, and confidence relative to conventional approaches is essential to validate the claimed advantage of this workflow.

3. Magnitude of improvement.

The most prominent experimental validation in Case Study 1 is an increase in the proportion of the major enantiomer in the product mixture by only 5.5% (from 87% to 92.5% when using L6 and L66, respectively). Although such improvements can be valuable in specific contexts, the result is not particularly compelling in the setting of high-throughput ligand screening to highlight the practical impact of a predictive model. In addition, the authors should report the effect of the change in ligand on the overall performance of the catalyst, for example by providing at least the TON and TOF of each catalyst.

4. Generality of the ligand improvement.

The authors claim that the model identifies ligands suited to distinct enantiodetermining steps based on the slightly superior performance of L66 versus L6 in the coupling of styrene oxide with 4-iodoanisole. However, the authors do not evaluate L66 in the original benchmark reaction (ethylene oxide with p-tolyliodide) or in any other reaction. Without this comparison, it remains unclear whether L66 is truly superior because it is tailored to the mechanistic shift in this reaction, and therefore the mechanism-informed descriptors captured the relevant mechanistic change, as claimed by the authors, or whether it is simply a generally slightly better ligand across several reactions.

(Remarks on code availability)

Version 1:

Reviewer comments:

Referee #1

(Remarks to the Author)

The authors did a very thorough job with detailed response and additional work done. This does make the manuscript significantly more robust. In the current form the manuscript is of very good quality and I have no concerns or further suggestions.

(Remarks on code availability)

Referee #2

(Remarks to the Author)

The authors have addressed all my concerns during the first round review. I recommend the acceptance of this work in Nature.

(Remarks on code availability)

Referee #3

(Remarks to the Author)

I co-reviewed this manuscript with one of the reviewers who provided the listed reports.

(Remarks on code availability)

Instructions available have improved significantly after revision, with a more detailed README file.

Referee #4

(Remarks to the Author)

The authors have not satisfactorily addressed my concerns.

1) Scale of demonstration. In my previous report, I raised concerns regarding the limited size of the experimental validation in Case Study 1, where only 16 ligands were evaluated. This number has not been increased in the revised version. While a new case study involving a larger virtual library has been added, this does not directly address the question of scalability for Case Study 1. If the approach is indeed as low-cost and scalable as claimed, expanding the experimental scope of Case Study 1 would substantially strengthen the manuscript and provide more convincing support for the authors' conclusions.

2) Benchmarking against conventional approaches. My concern regarding the absence of benchmarking against conventional methods in Case Studies 1 and 2 also remains. Although the newly added case study provides additional context, it does not remedy the lack of direct comparison for the original examples. Including such benchmarking for the primary case studies would greatly improve the credibility of the claims regarding efficiency and performance.

3) Magnitude of improvement. I would like to clarify that my concern is not with the formal relationship between ee/er and $\Delta\Delta G^\ddagger$, but rather with the synthetic significance of the reported improvements. While an increase from 87% to 92.5% ee is certainly not negligible, it is modest in view of the strong claims made about the impact of the methodology. Demonstrating the approach on a more challenging system lacking a known high-performing ligand, and achieving a more substantial improvement, would more convincingly illustrate the practical value of the method.

4) Generality of the ligand improvement. I remain uncertain about the interpretation of the results concerning ligand generality. In response to my request to test L66 in the original benchmark reaction (ethylene oxide with p-tolyliodide, compound 3), the revised manuscript includes new data; however, there appears to be an inconsistency between the structure shown (an ester) and the labeling of the compound as "3". In addition, the ee value reported for L6 does not match the previously reported value in Figure 2 (89%), which adds to the confusion. Moreover, the newly reported result for L66 in the reaction of compound 3 (92% ee) represents a 3% increase in ee relative to L6, which is consistent with L66 being a slightly better ligand in general, rather than one uniquely suited to a specific mechanistic scenario. This result weakens the claim that the model captures mechanistically specific ligand effects.

(Remarks on code availability)

Referee 1

[...] My recommendation would be to try to stress-test the approach and clearly identify the bounds of when it breaks. There are way too many papers in the field of ML for chemistry that over-promise and are not scrutinizing themselves enough.

Referee 1 makes a very important point regarding the extent of the ML models' domain of applicability and we share their concerns. We have therefore sought to push the bounds of our ensemble MLR modeling approach by evaluating its predictive accuracy on datasets involving either two completely unseen coupling partners or one electrophile class whose binding mode to the Ni center is substantially different, from an electronic and coordination perspective (see Fig. S12C). Indeed, in these situations we have found Model 2 from case study 2 to yield inaccurate predictions. We have added a new section (5.1 Out-of-sample predictions using Model 2) to the SI regarding reaction I (see Fig. S12), which we evaluated in the newly added case study 3:

Figure S12. **A.** Ni-catalyzed reductive arylation of benzylic acetals. **B.** Out-of-sample predictions using ensemble Model 2 from case study 2. **C.** Kernel density estimate of feature $d_{(Ni-C2s)}^{TSRE, Boltz}$ for reactions **D–G**, **H**, and **I**. The lowest energy TSRE conformer of **L1** for reaction **I** is shown.

In order to further investigate the domain of applicability of ensemble Model 2 from case study 2, we evaluated its ability to make accurate predictions on a reaction with another unseen $C(sp^3)$ -electrophile. We selected the coupling of benzaldehyde dimethylacetal and 4-Ac-phenyl iodide (Figure S12A, reaction I) to include one coupling partner from the original training set (aryl iodides, reactions **D–G**). However, while the $C(sp^3)$ -electrophiles in the training set (styrenyl aziridines and benzylic chlorides) possess a CH or CH_2 group α to the benzylic position, the $C(sp^3)$ -electrophile in reaction I possesses an ether substituent, which can coordinate η^2 to the Ni-center. We tested the model on results from 35 BiOx and 6 BiIm ligands reported in a group member's thesis in 2020 (see ref. 35 in the SI). Model 2 afforded inaccurate predictions on this reaction (MAE = 0.38 kcal/mol, $R^2 = 0.05$, Figure S12B), with the $\Delta\Delta G^\ddagger$ of some ligands being severely overpredicted and associated with a large degree of uncertainty (± 1.79 kcal/mol). To help understand potential reasons for the inaccurate predictions, we analyzed how the distribution of the features in the ensemble model (see Table S21) differs for reactions **D–G** vs. reaction **H** (for

which accurate out-of-sample predictions were obtained) and **I**. An example is given in Figure S12C, which shows the kernel density estimation curve of $d_{(\text{Ni}-\text{C}_{2\text{s}})}^{\text{TSRE, Boltz}}$ *i.e.*, the distance between Ni and the C(sp^3)-atom of the electrophile in the reductive elimination transition state. The distribution of this feature for reaction **H** more closely resembles that of reactions **D–G**, whereas for reaction **I** the median is shifted to lower values. This is consistent with the electrophile being more tightly bound to Ni owing to the presence of the O atom, which can act as an additional donor (see the lowest energy TSRE conformer of **L1** in Figure S12C).

Referee 2

1) While the Boruta algorithm and cross-validation are effective, incorporating more advanced feature selection techniques (e.g., SHAP analysis or graph-based descriptors) could further improve model interpretability and accuracy.

Referee 2 makes a valid point regarding the interpretability of machine learning models: specifically, nonlinear models (more so than multivariate linear regression models) may be hard to interpret. In the newly added case study 3 (see the detailed response to point 1 of Referee 4), we trained a Gaussian Process Regression (GPR) model to perform active learning with our EDBO+ platform. Therefore, following the Referee's suggestion, we performed SHAP analysis to identify the most important descriptors used by the GPR model (Fig. 6B). We also performed dimensionality reduction on this feature set to visualize the ligand search space (Fig. 6C).

2) Expanding beyond linear regression to include nonlinear methods (e.g., neural networks or kernel-based models) may better capture complex catalyst-substrate interactions, especially for highly flexible systems.

To address this comment, the following section (3.9 Non-linear regression) has been added to the SI:

To verify whether non-linear regression methods could better capture complex and dynamic catalyst–substrate interactions, we explored a variety of algorithms (random forest, gradient boosting, and neural networks) with the ROBERT package (<https://robert.readthedocs.io/en/latest/index.html>). Among the regression methods tested, GB (without low permutation feature importance filtering) was found to afford weak predictive accuracy (10×5-fold CV $R^2 = 0.51$, MAE = 0.22 kcal/mol, RMSE = 0.28 kcal/mol; Test $R^2 = 0.74$, MAE = 0.18 kcal/mol, RMSE = 0.22 kcal/mol, Figure S11). Additionally, the out-of-sample set (4 BiOx and 5 BiIm ligands tested to form products **16** or **23**) was incorrectly predicted ($R^2 = 0.01$, MAE = 0.63 kcal/mol, RMSE = 0.67, Figure S11), showing the inability of this non-linear method to predict untested ligand–substrate combinations with only sparse data available for training. The corresponding ROBERT report has been made available on GitHub.

Figure S11. ROBERT Results: GB, standard descriptor filter (no PFI). CV (Train + Validation) : Test = 79 : 21. Points (Train + Validation) : descriptors = 42 : 18.

3) The workflow assumes specific TSs (e.g., radical capture or reductive elimination) as enantiodetermining steps. If the authors could include alternative mechanistic pathways (e.g., concerted mechanisms), this could enhance robustness, particularly for controversial reactions.

Referee 2 raises an important concern regarding the featurization workflow. To address this, we have added the following section (2.6 Evaluation of alternative mechanistic pathways) to the SI:

In this study, the widely proposed stepwise mechanism for carbon–carbon bond formation was considered, leading to the generation and featurization of **TSRC**, **Int**, and **TSRE** structures. Recent work by Diao *et al.* suggested that an alternative concerted inner-sphere radical capture and carbon–carbon bond formation process may be viable. To assess the energetic feasibility of this mechanism, we reexamined the potential energy surface computations reported by Doyle *et al.* for the coupling of styrene oxides and aryl iodides. IRC computations were performed at the B3LYP/6-31G(d) level of theory starting from **TSRC**, leading to the optimization of the Ni(II)–benzylic radical complex. A relaxed PES scan (Figure S6) was conducted on this structure, shortening the C(*sp*²)–C(*sp*³) distance in order to look for a guess structure from which the concerted TS for radical capture and carbon–carbon bond formation could be optimized. However, during the scan the complex collapsed to the Ni(III) intermediate (*d* ~ 2.7 Å), and subsequent scan steps afforded the reductive elimination TS. Optimizing structures with *d* ~ 3.3 Å afforded **TSRC**. These results suggest that, for this reaction and at the level of theory reported by Doyle *et al.*, a concerted TS cannot be located. We note that in this work, the structures reported by Doyle *et al.* and by Reisman *et al.* were used as templates (Figure 2, Step 1a) because the PES for the enantiodetermining steps had been computed, with the assumption that these steps are relevant to the reactions investigated in case study 2. Applying the workflow outlined in Figure 2 to a different reaction class would require a preliminary computational investigation of the stereodetermining step(s).

Figure S6. Relaxed PES scan at the B3LYP/6-31G(d) level shortening the distance between the $C(sp^3)$ of the benzylic radical and the $C(sp^2)$ of the Ni(II)–aryl species. The electronic energy (in kcal/mol) relative to the Ni(I)–product complex is reported on the y -axis.

4) The authors are suggested to couple the computational predictions with automated high-throughput experimentation as it could accelerate iterative optimization and provide rapid feedback for model refinement.

A new case study (Case study 3: active learning with catalytically relevant descriptors) has been added to the manuscript (see the detailed response to point 1 of Referee 4). We used active learning to optimize an unreported asymmetric reaction, iteratively improving the GP regression model within our EDBO+ platform for enantioselectivity predictions. While our modeling approach could be coupled with HTE, only ~47% of the 195 ligands virtually screened are commercial, and accessing 104 non-commercial ones is synthetically intensive. Therefore, we posit that it is more time- and resource-efficient to perform high-throughput computations on this large virtual space and use an active learning approach that leverages fewer experimental data points (see Fig. 6).

Referee 3

1) It is important to include baseline models to the data presented, as they prove the modelling is challenging. The feature selection with the Boruta algorithm is valuable and a valuable approach for feature selection but it does not remove the need for a baseline model. Although results are promising, the presence of a validation strategy, including randomizing ee values within the CV folds, would be valuable and increase the reliability of this strategy.

Referee 3 correctly highlights the importance of baseline models and further validation strategies, such as randomization of the $\Delta\Delta G^\ddagger$ target values, to ensure the predictive accuracy of the MLR

models. To address these concerns, the following paragraph has been added to section 2.4 (Repeated, stratified, nested k -fold and 5×2 cross-validation schemes) of the SI:

Figure S5. Synoptic radar plot summarizing the performance of the **TSRE** MLR model in all the statistical tests.

To assess the capability of the repeated, stratified nested k -fold CV strategy (Figure S3) to identify MLR models able to predict ligands outperforming those in the training data, we used the COBRA web application (<https://www.aocdweb.com/OMtools/cobra>) to conduct a series of statistical tests on the MLR model using the **TSRE** features (*i.e.*, the top-performing model from the benchmarking study). These tests include leave-one-out CV, y -randomization, bootstrapping, predicting the top-performing ligands in the dataset, and simulating three cycles of catalyst optimization. While full details of these tests are given elsewhere (<https://www.aocdweb.com/OMtools/cobra/help>), the results are summarized in the synoptic radar plot shown in Figure S5 by visualizing deviations from the ideal behavior (*e.g.*, $R^2 = 1$). The radar plot indicates that the model passes all the tests, with small deviations from ideal behavior (compare Figure S5 with the prototype example reported by Cavallo *et al.*). In alignment with the finding by Cavallo *et al.*, y -randomization is the most problematic test for such a small dataset (29 datapoints), but the high value of the $1 - \text{MAE}_{T20}$, Pre_{80} , Acc_{80} , and Rec_{80} tests (0.72, 0.85, 0.91, 1.00, respectively) supports using this model to predict better-performing ligands.

2) On the code, some of the *.xlsx* files have different labels on the first row as what it seems to be expected from the code. Therefore, I would suggest reviewing the code provided on GitHub runs with the same files which are present in the same folder. An example run would also be helpful to include in the README.md file details on how others could run the code for one of the benchmarks, with the specific flags and file names. One example present is the inconsistency in the Ligand Class label keys that change from case study 1 to case study 2 (“Class” in “Case_Study_1_Linear_Random.xlsx”, to “Lig_Class” in “Case_Study_2_CH_Functionalization.xlsx”).

The .xlsx files on GitHub have been updated. Step 8 (Multivariate Linear Regression Modeling) in the README.md file on GitHub has been updated to provide a better guide on how the nested_CV.py script may be used.

3) Additionally, there are multiple new catalysts that are suggested by the model. A clarification on the reasons why **L65**, **L66**, and **L144** were selected from Table S12 would be valuable. Other ligands, such as **L73**, **L124**, and **L143** could be examples to be tested as they present promising $\Delta\Delta G^\ddagger$ values. There might be practical constraints other experimentalists might not be aware of and not encoded in the model (solubility or stability, to mention a few). The absence of these examples could lead to the reader interpreting it as cherry picking.

To clarify the choice of which ligands have been tested in case study 1, along with the results from the screening of the additional ligands suggested by Referee 3, the following section (3.3 Additional ligand screening data) has been added to the SI:

Among the ligands included in the “virtual” set, **L65**, **L66**, and **L144** were predicted by the ensemble models with linear or nonlinear parameters to afford the highest $\Delta\Delta G^\ddagger$ in the reaction to form product **23** (Table S12) and were tested first due to their availability in our lab. Following the identification of **L66** as a better-performing ligand, we purchased and evaluated three additional ligands (**L73**, **L124**, and **L143**) in the reaction to form product **23** that were also predicted to afford high $\Delta\Delta G^\ddagger$ values with an acceptable level of uncertainty. The results shown in Figure S7 indicate that **L66** remains optimal. The large deviation between predicted and measured $\Delta\Delta G^\ddagger$ value for **L73**, which afforded product **23** in only 2% ee, may be attributed to its inefficient ligation to the Ni-center due to the bulky *t*-butyl substituents. Additionally, we evaluated the performance of **L66** in the reactions yielding products **3** and **16** to verify whether it would afford a similar improvement in enantioselectivity compared to **L6** as it did for product **23** (Figure S8). **L6** and **L66** were found to afford similar ee values, suggesting that **L66** is tailored for forming product **23** in higher selectivity.

Figure S7. Additional ligands evaluated in the cross-coupling to form product **23**.

4) *Experimental data has been presented well. Although some chromatograms contain a large number of impurities, they were adequately excluded from model validation. The reason for the exclusions is also included in the SI, as low conversions will lead to poor measurements of enantiomeric excess. It would be appreciated if a clear threshold of conversion was explicitly included in the work.*

Referee 3 makes a valid point regarding the need to state a clear threshold for the exclusion of experimental results from the models' validation set. Indeed, we excluded from the out-of-sample sets ligands that were poorly soluble (**L28**, case study 1, as stated in Table S5) or whose catalytic turnover was < 1 (*i.e.*, yield $< 15\%$, **L57**, **L61**, **L63**, and **L68**, case study 2, as stated in Table S16). In order to clarify this further in the manuscript, the following sentences were added to the captions of Figure 4 and 5, respectively:

One poorly soluble ligand was excluded from the OOS set (see Table S5).

Ligands affording less than one catalyst turnover were excluded from the OOS set (see Table S16).

5) *Finally, a handful of validation experiments for the modeling workflow would be valuable. The current article does present true positives, however there is limited assessment of false negatives from the model predictions. A few entries of experimental screenings with **L1–L29** for substrates **16** or **23** (potentially randomly selected) could prove some insight into the false negatives the model might miss. Because these are expected to be negative results, not an extensive list of experiments should be expected but rather a minimum number to validate the mode's predictions and enhance robustness.*

True negatives (*i.e.*, ligands predicted and experimentally verified to afford $\Delta\Delta G^\ddagger < \mathbf{L6}$ for product **16** or **23**) were indeed included in case study 1. These reactions (see Tables S4 and S5) constitute the out-of-sample set shown in Fig. 4C. The ensemble MLR models of linear or SISSO-augmented parameters demonstrate good predictive accuracy on this set (MAE = 0.16 kcal/mol, $R^2 = 0.76$, and MAE = 0.11 kcal/mol, $R^2 = 0.60$, respectively). The models are therefore able to interpolate between the min and max $\Delta\Delta G^\ddagger$ values and make reliable predictions on untested ligand–substrate combinations (in the training data, every substrate combination is tested with **L6**, and every ligand is tested in the synthesis of product **3**).

The following sentences (lines 245–250 in the manuscript) have been updated for further clarity:

Before making predictions on additional ligands from a virtual library, we sought to experimentally validate the ensemble model. We tested 4 BiOx and 5 BiIm ligands in the cross-coupling to form **16** or **23** and, as expected, this resulted in low to mid *ee* values (39–71% *ee*, true negatives, see SI Tables S4 and S5), confirming the model's good out-of-sample predictive accuracy (MAE = 0.16 kcal/mol, RMSE = 0.18 kcal/mol, $R^2 = 0.76$, Fig. 4C).

Referee 4

1) Scale of demonstration: *A central selling point of the manuscript is scalability at low computational cost. Yet, in case study 1 they only evaluated 16 new ligands, which undermines their claim. If the methodology is indeed low-cost and scalable, one would expect to see hundreds*

of ligands virtually screened to showcase its full potential, or even an application to a novel reaction class beyond those already published by the same group. The current scope (two Ni-catalyzed case studies, both related to previously published work by the authors in JACS) feels insufficient to demonstrate broad applicability.

To address Referee 4's concerns about the scalability of the computational workflow and applicability to a novel reaction class, a new case study (Case study 3: active learning with catalytically relevant descriptors) has been added to the manuscript. Herein, we used published enantioselectivity data from 250 Ni-catalyzed cross-coupling reactions with *N,N*-ligands to pre-train a GPR model, which was employed for the optimization of an unpublished Ni-catalyzed reductive coupling. Conformational ensembles of **TSRC–Int–TSRE** of 195 ligands were featurized for this optimization campaign in order to demonstrate the full potential of the workflow to be scaled. In this work, each case study is in the context of Ni-catalyzed cross-coupling. This reaction class was strategically selected as a testing platform for the computational pipeline since distinct combinations of coupling partners necessitate different ligand classes for optimal performance. Furthermore, there is some understanding of common enantiodetermining steps (*i.e.*, **TSRC** and **TSRE**), which is needed for the featurization strategy.

Case study 3: active learning with catalytically relevant descriptors

Having validated our featurization strategy on retrospective studies, we next sought to apply it to the prospective optimization of an unreported asymmetric reaction, specifically the coupling of α -oxy radicals generated from benzylic acetals with aryl iodides (reaction **I**, Fig. 6A). We sought to: (1) explore a large virtual ligand space in order to demonstrate the scalability of the featurization workflow, and (2) identify promising ligands in a limited number of experiments. Since this C(*sp*³)-coupling partner was found to be outside the domain of applicability of ensemble Model 2 (see SI section 5.1 for full details), we integrated the mechanism-informed featurization strategy with our open-source active learning platform EDBO+ to leverage Bayesian theory for the optimization of enantioselectivity. Parameters of **TSRC**, **Int**, and **TSRE** for 195 chiral *N,N*-ligands in reaction **I** were collected, which corresponded to the optimization of ~70,000 geometries (completed in *ca.* 2 weeks). To avoid a combinatorial explosion of the search space, optimization efforts were focused on evaluating ligands for enantioselectivity, and therefore, reaction conditions (*e.g.*, solvent, reductant, temperature) were not varied (see SI section 5.3 for experimental details). Bayesian optimization is an ideal alternative to high-throughput screening in the context of asymmetric Ni-catalyzed cross-coupling reactions given the limited commercial availability of chiral ligands (only ~47% of the ligands virtually screened are commercial) and time associated with multi-step syntheses of non-commercial examples.

As demonstrated in case study 2, reactions involving distinct coupling partners may be simultaneously modelled using the mechanistically informed features. We therefore pre-trained a Gaussian Process Regression (GPR) model on **TSRC–Int–TSRE** descriptors of sparse data from published enantioselective Ni-catalyzed C(*sp*³)-cross-coupling reactions, maximizing information extracted from the literature (round 0, see SI section 5.2 for full details). This dataset is available on GitHub in the hope that it may help “cold start” the optimization of other transformations. At the beginning of the active learning campaign (round 0), benzylic acetals are entirely unseen coupling partners; the GPR model is progressively retrained on small batches of experiments (4-6 ligands screened per round) until the expected improvement (EI) of $\Delta\Delta G^\ddagger$ indicates diminishing

return to performing additional experiments (avg. EI \sim 5% for round 5, Fig. 6A) and the model's prediction accuracy is acceptable (MAE = 0.33 kcal/mol at round 4, see Fig. S13).

Despite the modest enantioselectivity achieved in reaction I (max $\Delta\Delta G^\ddagger = 0.98$ kcal/mol, 68% ee), EDBO+ identified the likely optimal ligands in this search space within only 4 rounds *i.e.*, 26 experiments. Notably, the top performing ligand class *i.e.*, PyOx (see ligands **L91**, **L79**, and **L116** in Fig. 6A) is different from the optimal classes for the literature reactions used to pre-train the GPR model (where BiIm, BiOx, PyIm, or BOX ligands are top-performing). To determine the most influential descriptors in the GPR model, we performed SHAP analysis (Fig. 6B), which identified $\%V_{\text{bur}}$ parameters and Hirshfeld charges as the most significant contributors to the model's outcome; 36% of the most important features are extracted from **TSRE**, followed by **Int** (34%), and **TSRC** (30%). We then reduced this subset of the feature space to two dimensions using the Uniform Manifold Approximation and Projection (UMAP) method, revealing how distinct ligand classes are clustered together (Fig. 6C). During the optimization campaign, EDBO+ explores and exploits these different ligand types *e.g.*, BOX in round 2 and PyOx in rounds 3-4, identifying ligands from the PyOx class **L116** (and **L79**) as optimal. By integrating the mechanism-informed featurization strategy with an existing data science-guided reaction optimization platform, we were able to leverage literature data and rapidly identify a selective ligand class for a novel asymmetric cross-coupling reaction.

A. Rapid identification of selective ligands using Bayesian optimization

B. Feature importance using SHAP analysis

C. Dimensionality reduction of virtual ligand space

Fig. 6. A, Prospective application of the featurization strategy: Ni-catalyzed coupling of benzylic acetals and aryl iodides. The $C(sp^3)$ -coupling partner is unseen (*i.e.*, missing from the training data). The GPR model is trained on literature data prior to the start of the active learning campaign using EDBO+. In the plot, $\max \Delta\Delta G^\ddagger$ indicates the maximum enantioselectivity achieved up to round n . **B**, Beeswarm plot highlighting the 10 most important features contributing to the model's regression, ordered according to their mean absolute SHAP value. **C**, UMAP chemical space representation of the ligand space for Reaction **I**. Dimensionality reduction was performed using the top-ranked features whose mean absolute SHAP contribution to $\Delta\Delta G^\ddagger$ was ≥ 0.10 kcal/mol, retaining features in descending importance until they collectively accounted for $\geq 90\%$ of the total mean absolute SHAP importance.

2) *Benchmarking against conventional approaches*: The efficiency of the method is framed by comparing “3 experiments (vs. 45 virtually screened).” However, no direct benchmark is provided against brute-force or high-throughput experimental screening, which would produce a larger number of results and, more importantly, deliver definitive outcomes without uncertainty. A quantitative comparison of time, cost, and confidence relative to conventional approaches is essential to validate the claimed advantage of this workflow.

In the newly added case study (Case study 3: active learning with catalytically relevant descriptors), only ~47% of the 195 ligands virtually screened are commercial, at an estimated cost of ~\$5,000 (<https://www.molport.com/shop/swl-step-1>, 91 ligands, 50-100 mg each). Using an active learning platform, only 2 non-commercial ligands (**L118** and **L120**) had to be synthesized, each *via* a telescoped four-step procedure, which took approximately one week. Although we cannot reliably quantify how long it would take to synthesize the remaining 102 ligands, screening this entire space of 195 catalysts *via* HTE would be highly impractical. Active learning is therefore an ideal alternative to HTE, as the requirement to generate a large number of experimental observations is minimized. Furthermore, none of the remaining 102 ligands were predicted by the GPR model (round 5) to afford an increase in enantioselectivity compared to **L116**.

3) *Magnitude of improvement*: The most prominent experimental validation in case study 1 is an increase in the proportion of the major enantiomer in the product mixture by only 5.5% (from 87% to 92.5% [*er*] when using **L6** and **L66**, respectively). Although such improvements can be valuable in specific contexts, the result is not particularly compelling in the setting of high-throughput ligand screening to highlight the practical impact of a predictive model. In addition, the authors should report the effect of the change in ligand on the overall performance of the catalyst, for example by providing at least the TON and TOF of each catalyst.

When assessing the improvement in enantioselectivity achieved in case study 1, it is important to consider the non-linear relationship between enantiomeric excess (*ee*)/enantiomeric ratio (*er*) and $\Delta\Delta G^\ddagger$. The increase in $\Delta\Delta G^\ddagger$ achieved with **L66** vs. **L6** for product **23** is 0.34 kcal/mol, an extrapolation equal to twice the OOS MAE of the ensemble MLR model with linear parameters (0.16 kcal/mol). To screen the virtual set (Figure S10), the ensemble models are making predictions on combinations of entirely unseen ligands and products that have been seen only once in the training set. Therefore, the identification of **L66** represents a valuable validation of the computational workflow. In this manuscript, regression models were trained solely on enantioselectivity data and therefore other catalyst performance parameters (*e.g.*, TON, or TOF)

were not optimized. Isolated or assay yields of each reaction are reported in the SI to reflect the overall catalyst performance for each ligand.

4) *Generality of the ligand improvement:* The authors claim that the model identifies ligands suited to distinct enantiodetermining steps based on the slightly superior performance of **L66** versus **L6** in the coupling of styrene oxide with 4-iodoanisole. However, the authors do not evaluate **L66** in the original benchmark reaction (ethylene oxide with *p*-tolyl iodide) or in any other reaction. Without this comparison, it remains unclear whether **L66** is truly superior because it is tailored to the mechanistic shift in this reaction, and therefore the mechanism-informed descriptors captured the relevant mechanistic change, as claimed by the authors, or whether it is simply a generally slightly better ligand across several reactions.

To address Referee's 4 questions on the generality of the ligand improvement, additional experiments with **L66** were conducted and the following section (3.3 Additional ligand screening data) has been added to the SI:

[...] Additionally, we evaluated the performance of **L66** in the reactions yielding products **3** and **16** to verify whether it would afford a similar improvement in enantioselectivity compared to **L6** as it did for product **23** (Figure S8). **L6** and **L66** were found to afford similar *ee* values, suggesting that **L66** is tailored for forming product **23** in higher selectivity.

Figure S8. Evaluation of **L66** in cross-couplings to form products **3** and **16**.

Dear Editor,

Thank you for giving us the opportunity to address Referee 4's comments and suggestions. We are happy to see that Referees 1–3 are satisfied with the edits and believe the manuscript to be suitable for publication in Nature. As you suggested, we would like to respond to and clarify the points raised by Referee #4 in order to verify if additional experiments are indeed needed, and if so to help make plans for any further revisions.

In their first comment, Referee #4 once again expressed concerns over the scalability of our approach, specifically regarding the size of the experimental validation of Case Study 1, mentioning that only 16 ligands were evaluated. To validate the ensemble models' predictions, we experimentally tested 9 ligands in the cross-coupling reaction to form either product 16 or 23 (Fig. 4C). Subsequently, we tested an additional 6 ligands in the reaction to form product 23 (Fig. 4D and Fig. S7). The number of reactions experimentally tested is only a fraction of the substrate–ligand combinations virtually screened:

- For both products 16 and 23, we have extracted features from TSRC, Int, and TSRE for 29 ligands (L1–L29), as well as the 16 ligands in the “virtual set” (Figure S10). This corresponded to the generation and optimization of 128,488 structures.
- Not reported is that we have also virtually screened L1–L29 against four additional products (15, 17, 22, and 26 from the original publication, *J. Am. Chem. Soc.* 2021, 143, 15873–15881). This corresponded to the optimization and featurization of an additional 144,605 TSRC, Int, and TSRE number of geometries. We have not included this data in the manuscript as none of these ligands were predicted to afford higher enantioselectivity for products 15, 17, 22, and 26 than L6. If you (and the Referee) think this will further demonstrate scalability of the approach, we will be happy to share this information in the Supporting Information.

Furthermore, regarding Case Study 2, for each of the four reactions D–G, we have featurized TSRC, Int, and TSRE for all the combinations of one reaction's product with the ligands tested in the other three reactions (i.e., 16 out-of-sample ligands for reaction D, 25 for reaction E, 31 ligands for reaction F, and 29 for reaction G). This required the optimization of 159,417 geometries. We only reported a subset of this data in the manuscript (14 out-of-sample D and G reactions experimentally tested to validate Model 2, Tables S16 and S17), but we are again happy to share these additional computational results.

We believe these numbers (along with the ~70,000 geometries optimized for Case Study 3 in ca. 2 weeks) support the scalability of the computational protocol. Reading Referee #4's latest comment, we are unsure how expanding the experimental scope of Case Study 1 would further support the demonstration of the scalability of the computational protocol. If needed, there are two courses of action we could explore:

- 1) We have access to an additional 28 BiIm/PyOx ligands in our chemical inventory. We could expand the “virtual set” for products 16 and 23 to include these ligands (and optimize their corresponding TSRC–Int–TSRE structures).

- 2) If Referee #4 is concerned with respect to the number of substrates included in the experimental validation set, we could experimentally test additional substrate–ligand combinations. However, we are unsure how much value additional out-of-sample test data points would add if none are predicted to be a significant improvement over previously reported enantioselectivities (see *J. Am. Chem. Soc.* 2021, 143, 15873–15881).

Regarding Referee #4's second comment, we are unsure what "conventional methods" for benchmarking our modeling workflow in Case Study 1 would be. The premise of Case Study 1 is that, conventionally, a set of chiral ligands is evaluated when optimizing one specific ("model") substrate combination, under the assumption that the optimal ligand would then afford higher enantioselectivities when modifications are made to the substrates. In their first report, Referee #4 mentioned "benchmark[ing] against brute-force or high-throughput experimental screening". While one could imagine exhaustively evaluating the entire substrate–ligand matrix, it is extremely difficult to test multi-substrate, multi-ligand combinations when the optimization goal is enantioselectivity, as the product of each experiment needs to be isolated for evaluation of enantiomeric excess using standard analytical techniques (chiral HPLC or SFC). Jacobsen and coworkers (*Nature* 2022, 610, 680–686) have reported the development of advanced analytical techniques in combination with HTE to enable the exhaustive screening of a substrate–catalyst matrix. The impact of this study, published in *Nature*, underscores the challenges associated with obtaining the enantioselectivity data which would be required for direct benchmarking of our workflow, and we do not have the chiral SFC-MS technology available in our labs to do this.

With respect to Referee #4's third comment, the first case study sought to demonstrate that the featurization strategy can capture substrate–catalyst interactions in models trained on sparse data. The most compelling aspect of this case study, in our opinion, is that we were able to train a model on limited "L-shaped" data (i.e., one substrate combination \times 29 ligands and one ligand \times 23 substrate combinations) and validate this model using unseen substrate–ligand combinations and unseen ligands. The virtual screening approach taken (Fig. 4D) was intended to serve as a practical application of this strategy for optimization of specific substrates of interest, but we agree that the reported increase in enantioselectivity for product 23 from 74% ee to 85% ee (or 87:13 er to 92.5:7.5 er) is modest, likely because this is a published reaction that has already been optimized to provide moderate to high enantioselectivity for all products reported, as noted by the Referee. Regarding Referee #4's request for a more dramatic example, we believe Case Studies 2 and 3 showcase more dramatic examples of the featurization strategy by applying the workflow to predictions on unseen coupling partners, ligands, and reaction classes. We will adjust language in Case Study 1 to ensure the main text clearly represents this data with accuracy.

We are appreciative of Referee #4's fourth comment and apologize for the confusion. The product shown in the main text (ethylene oxide with p-tolyl iodide) was incorrectly labeled as 3 in Fig. 4B and should be labeled as product 6. This substrate combination was used as the model system for evaluating L1–L29 in the original report and was reported in 89% ee on a 0.05 mmol scale and 90% ee on a 0.5 mmol scale. To address the Referee's request to evaluate the generality of L66, we evaluated L66 in cross-couplings to form products 3 and 16 from the original report, as we were able to easily characterize and quantify enantiomeric excess for these two products. The results suggest that L66 is equally as competent as L6 in these reactions, but L66 did not provide an increase in enantioselectivity for either product 3 (91% ee with L6 vs. 92% ee with L66) or product

16 (78% ee with L6 vs. 79% ee with L66) (Figure S8). We apologize for this error and will be sure to correct the compound numbering in both the main text and Figure 4.

Overall, we do think that the indicated additional data that we already have could address Referee #4's main concerns in comment 1. That said, we would gladly perform any additional computations or experiments needed to strengthen the quality of our manuscript. Before doing so, we hope to clarify what the most effective way of addressing their first two points would be.